# Enhance Multi-View Classification through Multi-Scale Alignment and Expanded Boundary

**Yuena Lin** [1,†], **Yiyuan Wang** [1,2,†], **Gengyu Lyu** [1,*], **Yongjian Deng** [1],
**Haichun Cai** [3], **Huibin Lin** [3], **Haobo Wang** [4], **& Zhen Yang** [1,*]

[1] College of Computer Science, Beijing University of Technology, Beijing

[2] Idealism Beijing Technology Co., Ltd., Beijing

[3] College of Computer and Data Science, Fuzhou University, Fuzhou

[4] School of Software Technology, Zhejiang University, Hangzhou

`yuenalin@126.com, yiyuanwang@emails.bjut.edu.cn`
`lyugengyu@gmail.com, fjsmchc@163.com, 221010005@fzu.edu.cn`
`wanghaobo@zju.edu.cn, {yjdeng, yangzhen}@bjut.edu.cn,`

## Abstract

Multi-view classification aims at unifying the data from multiple views to complementarily enhance the classification performance. Unfortunately, two major problems in multi-view data are damaging model performance. The first is **feature heterogeneity**, which makes it hard to fuse features from different views. Considering this, we introduce a multi-scale alignment module, including an instance-scale alignment module and a prototype-scale alignment module to mine the commonality from an inter-view perspective and an inter-class perspective respectively, jointly alleviating feature heterogeneity. The second is **information redundancy** which easily incurs ambiguous data to blur class boundaries and impair model generalization. Therefore, we propose a novel expanded boundary by extending the original class boundary with fuzzy set theory, which adaptively adjusts the boundary to fit ambiguous data. By integrating the expanded boundary into the prototype-scale alignment module, our model further tightens the produced representations and reduces boundary ambiguity. Additionally, compared with the original class boundary, the expanded boundary preserves more margins for classifying unseen data, which guarantees the model generalization. Extensive experiment results across various datasets demonstrate the superiority of the proposed model against existing state-of-the-art methods.

## 1 Introduction

Multi-view data are widely applied in real-world applications by combining diverse heterogeneous features to describe the same object (Wang et al. (2023a); Zhang et al. (2024b)). Unluckily, **feature heterogeneity** and **information redundancy** concealed in the multi-view data have become the obstacles in multi-view learning (MVL) (Tan et al. (2024)). The feature heterogeneity means that the multi-view data are collected from different data distributions without a unified data format, causing fusion difficulty among the view features (Wang et al. (2024a)). The information redundancy signifies the latency of unnecessary features or ambiguous data, which hinders the ability to delimit the decision boundaries appropriately and naturally engenders classification ambiguity.

Existing MVL models mainly delve into alleviating the heterogeneity by finding informative common subspaces (Zhang et al. (2023a)) or ideal view weights (Hu et al. (2022)). The former explores the feature consistency and complementarity among the multi-view data (Lyu et al. (2024b); Wang et al. (2024c); Liang et al. (2024)), which neglects significant class differences for separating reliable decision boundaries. The latter presumes that the qualities or importances of views are stable for samples (Kumar & Maji (2023)), and aims at allocating appropriate weights to different views (Houfar et al. (2023); Liu et al. (2023)). However, the varied view qualities fail to match diverse samples practically (Han et al. (2022b)), which also leads to unreliable decision boundaries. For instance,

---

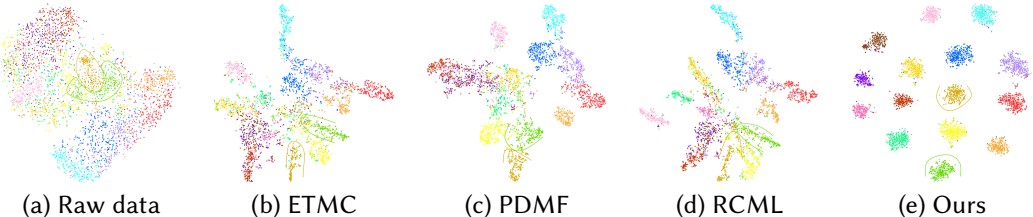

(a) Raw data     (b) ETMC     (c) PDMF     (d) RCML     (e) Ours

Figure 1: t-SNE on Scene15 dataset. The boundaries of the green class and the brown class are delineated. Compared with existing advanced models, our model shows much more distinct boundaries.

tigers and zebras have similar stripes but different colors, which makes the model give the color view a higher weight. Nevertheless, zebras and pandas have the same colors but disparate stripes, which induces the model to weigh more for the stripe view. Such scenarios easily result in unstable qualities or significances of different views to construct unreliable decision boundaries eventually.

Recent MVL works are dedicated to realizing trusted decisions by estimating the inherent uncertainties among multi-view data (Xu et al. (2023b; 2024)), and integrating multiple views at an evidence level. Actually, though they modify the logits to realize better classification decisions (Sensoy et al. (2018)), they still fail to deal with the aforementioned problems. To illustrate the issue and our motivation intuitively, we provide the t-SNE in Figure 1, which includes our model MAMC (enhance multi-view classification through multi-scale alignment and expanded boundary) and three advanced MVL models ETMC (Han et al. (2022b)), PDMF (Xu et al. (2023b)), and RCML (Xu et al. (2024)) on Scene15 dataset. The t-SNE reflects two issues of current MVL models: 1) They cannot well mine the commonality to tighten the representations; 2) They ignore to exploit the class difference for clearing the ambiguous decision boundaries. These observations imply that ***they are unable to reduce the feature heterogeneity and the information redundancy satisfactorily***. Motivated by these observations, *our model focuses on **mining commonalities** among view representations and **learning clear decision boundaries** in the representation space simultaneously to tackle the feature heterogeneity and the information redundancy.* On the one hand, MAMC introduces a multi-scale alignment module to learn expressive view representations by mining the commonalities among views and class differences among prototypes; on the other hand, it expands the original class boundaries to the wider decision boundaries in the representation space, fitting ambiguous samples to clear the blurry areas between different classes adaptively. Specifically, the multi-scale alignment module includes two complementary contents: 1) An instance-scale alignment module mines the inter-view commonality by aligning the view representations of each instance. 2) A prototype-scale alignment module exploits the inter-class difference from the instances with the same labels to find appropriate decision boundaries jointly.

The design of expanded boundaries is based on fuzzy set theory (Zhang et al. (2023b)), and its expansion process involves fuzzy representations and crisp representations, where the fuzzy representations are defined in the fuzzy representation space while the crisp representations are in the normal representation space. The core of boundary expansion is based on the assumption that crisp representations are a special type of fuzzy representations. It enables us to find learnable positives of the crisp representations in the fuzzy representation space, which are then employed to construct elements of the expanded boundary. Finally, the expanded decision boundary is integrated into the prototype-scale alignment process. By narrowing the expanded boundary and its corresponding class prototype while distancing other class prototypes, the inter-class difference is sufficiently exploited. Additionally, the expanded boundary reserves extra space for instances that belong to the class latently, which is beneficial for improving the model generalization. Our contributions are summarized as follows:

- We propose MAMC to address the problems caused by the feature heterogeneity and the information redundancy. Through introducing the multi-scale alignment module, MAMC sufficiently mines the inter-view commonality to tighten instances and inter-class difference to delimit the decision boundaries.

- We propose a novel self-adaptive expanded boundary to tackle the ambiguous decision boundaries. By integrating the expanded boundary into the prototype-scale alignment module, the inter-class difference is reinforced to clear the ambiguous decision boundaries, while the representations inside the boundary are tightened.

- We explain the rationality of the model design theoretically, and extensive experimental results across diverse public datasets and comprehensive experimental analysis have

verified that the proposed model shows significant superiority against the existing state-of-the-art models.

## 2 Preliminaries

### 2.1 Fuzzy Set Theory

Fuzzy set theory builds on classical (crisp) set theory by introducing the concept of membership degrees, which allows for fine-grained definitions of whether an element belongs to a set (Zadeh (1965)). A fuzzy set $\mathcal{A}$ in the universe $\mathbb{U}$ is characterized by a membership function $f_{\mathcal{A}}(x)$ which allocates each element in $\mathbb{U}$ with a real number in $[0, 1]$, where $\mathbb{U}$ is defined as a set that contains all the elements under consideration for a particular discussion or problem, and $f_{\mathcal{A}}(x)$ represents the membership degree of $x$ in $\mathcal{A}$ (Zhelezniak et al. (2019)). Fuzzy set theory provides a powerful framework for reasoning about sets with uncertainty, but the specification of membership functions depends greatly on domain knowledge. Specifically, we have the following definitions:

**Definition**: A function $f_{\mathcal{A}} : \mathbb{U} \to \mathbb{L} \subseteq \mathbb{R}$ is called a membership function.

**Definition**: A tuple $\mathcal{A} = (\mathbb{U}, \ f_{\mathcal{A}})$ including a universe $\mathbb{U}$ and a membership function $f_{\mathcal{A}}$ is called a fuzzy set.

Following the definitions, when $\mathbb{L} = [0, 1]$, $\mathcal{A}$ is a fuzzy set. When the membership function takes on only two values 0 and 1, namely $\mathbb{L} = \{0, 1\}$ according to whether $x$ belongs to $\mathcal{A}$. In this case, $\mathcal{A}$ reduces to a classical set.

### 2.2 Multi-view Learning

Joint learning on multi-view data has been proven effective on various tasks, such as clustering (Zhuge et al. (2020); Shen et al. (2024a)), and classification (Jiang et al. (2024); Lyu et al. (2024a)). Current works are roughly divided into the subspace-based (Shen et al. (2024b)) and view-weighted approaches (Zhang et al. (2024a)). The first category aims at finding a common subspace where the consensus representations are generated for all the views Wang et al. (2023b). Canonical correlation analysis (CCA) based approaches are one type of representative methods that project different views into a subspace by maximally correlating them. (Kumar & Maji (2023)) proposed D2CCA that integrates the CCA theory with the objective function to find the joint representations. (Yuan et al. (2022)) proposed a canonical $\mathcal{F}$-correlation framework where each feature was projected into a certain space by an arbitrary nonlinear mapping. Meanwhile, many works are dedicated to searching optimal weights for different views. (Xu et al. (2020)) proposed to make deep interactive information using view-specific information in an adaptive weighted manner, and seamlessly integrated the view-specific information with a multi-view loss fusion strategy to achieve joint decisions. (Houfar et al. (2023)) attended the distinctions between different views because of the sample importances, and proposed a dynamic learning strategy for automatically weighting views and samples. However, the subspace-based approaches mainly focus on mining feature commonality to handle feature heterogeneity but ignore the inter-class difference for delimiting the decision boundaries. And the view-weighted approaches have trouble producing generalized weights for views, which also makes it hard to find an effective boundary. In summary, these approaches fail to tackle the problem caused by information redundancy, especially when ambiguous samples are distributed around the decision boundaries. Motivated by this, we focus on mining commonalities among view representations and learning clear decision boundaries in the representation space simultaneously to alleviate the feature heterogeneity and the information redundancy.

### 2.3 Multi-view Classification

Existing multi-view classification methods can generally be divided into two categories: feature fusion-based and decision-based methods. Feature fusion-based methods focus on effectively combining features from all views (Meng et al. (2024)). MV-HFMD (Black & Souvenir (2024)) explores feature fusion by introducing a novel fusion scheme and mutual distillation to adapt neural networks for multi-view classification. Mmdynamics (Han et al. (2022a)) dynamically assesses the informativeness of both feature-level and view-level data across different samples, ensuring trustworthy integration of multiple views. However, these methods often neglect feature heterogene-

ity, which undermines the effectiveness of feature fusion. For the decision-based methods, they are dedicated to making more reasonable decisions. ETMC (Han et al. (2022b)) introduces a new paradigm for multi-view learning by dynamically integrating different views at the evidence level, thereby enhancing classification reliability. RCML (Xu et al. (2024)) attends to that conflictive information in different views, and proposes a conflictive opinion aggregation strategy that exactly models the relation of multi-view common and view-specific reliabilities. MVAKNN (Fan et al. (2023)) utilizes the Dempster–Shafer theory to integrate information from each view, successfully extending adaptive KNN to a multi-view setting. IPMVSC (Hu et al. (2023)) constructs a multi-view fuzzy classification model inherited from the natural interpretability of fuzzy rule-based systems to realize interpretable classification. Many methods of this kind interact with the multiple views only in the decision phase, which cannot well integrate the information from all the views. In contrast, our model aligns view features during the representation learning phase and combines them in the decision phase, allowing for more effective and interactive information integration across all views.

## 3 Proposed Method

The architecture of our model is shown in Figure 2, including view-specific representation extractors, a multi-scale alignment module, and a joint classifier. To detail the model architecture conveniently, we first provide some notations.

**Notations**. Formally, a multi-view dataset $\mathcal{D} = \{(\{\boldsymbol{x}_i^{(v)}\}_{v=1}^V, y_i)|1 \leq i \leq N\}$ is defined as $N$ *i.i.d.* instances with $V$ views and corresponding labels. We use a real-value vector $\boldsymbol{x}_i^{(v)} \in \mathbb{R}^{D_v}$ $(1 \leq v \leq V, i = 1, \cdots, N)$ to denote the feature vector of $i$-th instance for the $v$-th view, where $D_v$ is the feature dimension of the $v$-th view and $K$ is the number of classes. For the labels, we use a natural number $y_i \in \mathbb{N}$ to represent the ground truth of $i$-th instance and $\hat{\boldsymbol{y}}_i \in \{0, 1\}^K$ to represent the corresponding one-hot vector. To integrate information from the multiple views, our model aims at learning a desired mapping $f \colon \mathcal{X} \to \mathcal{Y}$ from the multi-view dataset $\mathcal{D}$ for better predicting the correct label of unseen instances.

### 3.1 View-Specific Representation Learning

Heterogeneous multi-view data conceal many inherent issues, such as inconsistent dimensions of view features. To address the issues, we employ multiple view-specific auto-encoders to generate exclusive view representations in the same dimension (Zhong et al. (2024); Jiang et al. (2024)). For the $v$-th view data $\boldsymbol{X}^{(v)}$, we apply a view-specific encoder $E^{(v)}$ to produce the view representation matrix $\boldsymbol{H}^{(v)}$ and a decoder $D^{(v)}$ to restore the raw view feature, where the encoder and the decoder are constructed by multi-layer perceptions (MLPs). The reconstruction process is given by

$$\mathcal{L}_{rec} = \frac{1}{V} \sum_{v=1}^V \left\| \boldsymbol{X}^{(v)} - D^{(v)} \left( E^{(v)}(\boldsymbol{X}^{(v)}) \right) \right\|_F^2. \tag{1}$$

### 3.2 Instance-Scale Alignment

The key to reducing the feature heterogeneity of multi-view data is to mine the inter-view commonality. Contrastive learning exploits commonality among data by pulling together the positives while pushing away the negatives (Wang et al. (2024b)). The instance-scale alignment module combines contrastive learning and regards all the view representations of each instance as positives while others as negatives. Concretely, the view representations $\{\boldsymbol{h}_i^{(v)}\}_{v=1}^V$ abstracted from the view-specific encoders are recognized as positives and others are negatives. Hence, the instance-scale alignment loss is

$$\mathcal{L}_{ins} = -\frac{1}{NV} \sum_{v=1}^V \sum_{u \neq v} \sum_{i=1}^N \log \frac{e^{-Dis\left(\boldsymbol{h}_i^{(v)}, \boldsymbol{h}_i^{(u)}\right) \cdot \tau_{ins}}}{e^{-Dis\left(\boldsymbol{h}_i^{(v)}, \boldsymbol{h}_i^{(u)}\right) \cdot \tau_{ins}} + \sum_{r=u,v} \sum_{j \neq i} e^{-Dis\left(\boldsymbol{h}_i^{(v)}, \boldsymbol{h}_j^{(r)}\right) \cdot \tau_{ins}}}, \tag{2}$$

where $\tau_{ins}$ is a temperature coefficient, and $Dis\left(\cdot, \cdot\right)$ is a distance function. By aligning the positive view representations, the critical common semantics to recognize similar samples are exploited,

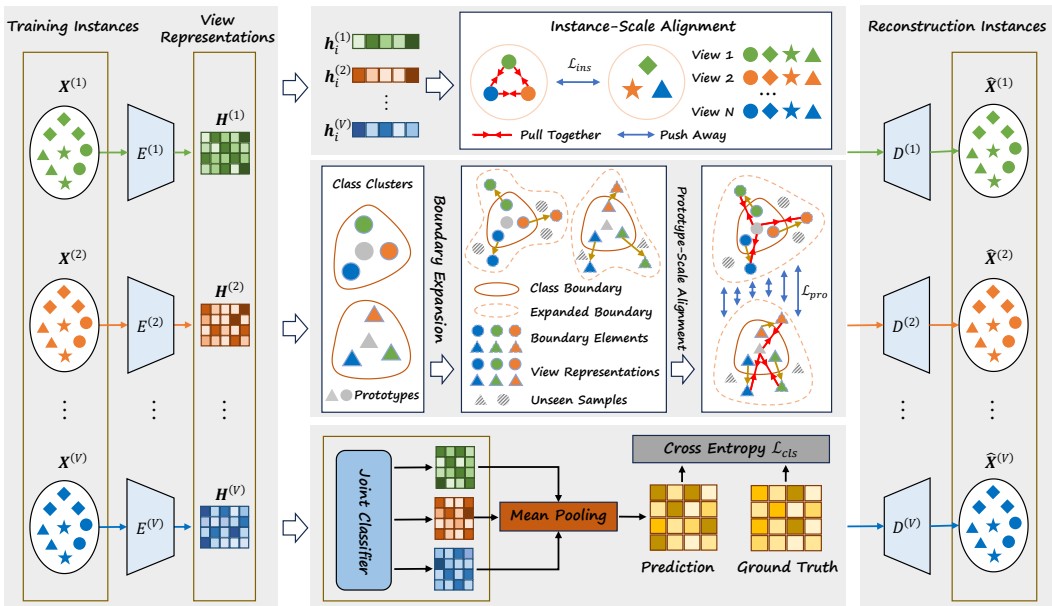

Figure 2: Three main components of the proposed model: (1) View-specific representation extractors. Multiple view-specific auto-encoders abstract critical view representations $\{\boldsymbol{H}^{(v)}\}_{v=1}^{V}$ from multi-view data $\{\boldsymbol{X}^{(v)}\}_{v=1}^{V}$. (2) Multi-scale alignment module. The instance-scale alignment module aligns view representations $\{\boldsymbol{h}_i^{(v)}\}_{v=1}^{V}$ of each instance $\boldsymbol{x}_i$, where the icons with various shapes correspond to different instances and those with different colors correspond to representations from different views. The prototype-scale alignment module integrates the adaptive expanded boundaries, and then aligns the view representations with the same labels. (3) Joint classifier. All the view representations are sent into the joint classifier to obtain fused predictions.

while the feature heterogeneity is mitigated. By differing the negatives, the discrepancies between different samples gradually emerge, thereby making the learned representations more suitable for delimiting the decision boundaries.

## 3.3 PROTOTYPE-SCALE ALIGNMENT

In contrast to the instance-scale alignment module, the goal of the prototype-scale alignment module is to exploit the inter-class difference for delimiting clear decision boundaries. The core of achieving this goal is to construct the expanded boundary.

### 3.3.1 EXPANDED BOUNDARY

Since the information redundancy in multi-view data easily blurs the decision boundaries, general prototype learning ignoring the inter-class difference has limited ability to work effectively. Instead, we propose an adaptive expanded boundary to assist the prototype-scale alignment module, expecting to clear the ambiguous boundaries.

To sufficiently gather the class information for exploiting the inter-class difference, we collect all the view representations with the same labels to construct the class prototypes and use them to centralize the view representations. For the training data of the $k$-th class, the centralizing process is defined as:

$$\boldsymbol{c}_k = \frac{1}{V\,|Y_k|} \sum_{v=1}^{V} \sum_{y_i=k} \boldsymbol{h}_i^{(v)}, \tag{3}$$

$$\boldsymbol{z}_i^{(v)} = \boldsymbol{h}_i^{(v)} - \boldsymbol{c}_k, \ v = 1, \cdots, V, \tag{4}$$

where $\boldsymbol{z}_i^{(v)}$ is the centralized representation vector of the $i$-th instance in the $v$-th view, $Y_k$ is the set including the training data whose labels are $k$, and $|\cdot|$ is used for counting the cardinality of a set. $\boldsymbol{c}_k$ is the prototype vector of the $k$-th class, which filters the high-frequency noise by averaging.

A key step of boundary expansion is to construct the fuzzy representation space with membership functions, and the Gaussian membership function is the most universal one (Li et al. (2023)). The appropriate number of membership functions imparts the expanded boundary with more adaptability. To this end, we adopt $L$ Gaussian membership functions. For the $l$-th membership function, the corresponding membership degree is defined by

$$\boldsymbol{r}_{i,j,l}^{(v)} = \exp\left\{-\frac{\left(\boldsymbol{z}_{i,j}^{(v)} - \boldsymbol{m}_{j,l}\right)^2}{2\boldsymbol{\delta}_{j,l}^2}\right\}, \; l = 1, 2, \cdots, L, \; j = 1, 2, \cdots, d, \tag{5}$$

where $d$ is the dimension of the view representation, $\boldsymbol{z}_{i,j}^{(v)}$ is the $j$-th element of $i$-th instance in the $v$-th centralized view representation vector, $\boldsymbol{m}_{j,l}$ and $\boldsymbol{\delta}_{j,l}$ are the trainable mean and standard deviation of the $l$-th Gaussian membership function respectively. These membership degrees are fused to obtain the final one for producing fuzzy representations, and the process is defined as

$$\tilde{\boldsymbol{r}}_{i,j}^{(v)} = Comb\left\{\boldsymbol{r}_{i,j,l}^{(v)}\right\}, \; l = 1, 2, \cdots, L, \tag{6}$$

$$\widetilde{\boldsymbol{Z}}^{(v)} = \widetilde{\boldsymbol{R}}^{(v)} \odot \boldsymbol{Z}^{(v)}, \tag{7}$$

where $\tilde{\boldsymbol{r}}_{i,j}^{(v)}$ is the element of the membership degree matrix $\widetilde{\boldsymbol{R}}^{(v)}$, and $\odot$ is Hadamard product to transform the crisp view representation $\boldsymbol{Z}^{(v)}$ into the fuzzy representation $\widetilde{\boldsymbol{Z}}^{(v)}$. $Comb$ is the combination operator to fuse membership degrees and how to determine it will be explained later.

According to properties of membership functions, each value of $\widetilde{\boldsymbol{R}}^{(v)}$ is limited in $[0, 1]$, which implies that the crisp representations can be viewed as one special case of fuzzy ones. Based on this, we regard crisp representations and fuzzy representations as positives. Besides, we assume that their discrepancies reflected by the coordinate differences are tolerable since the discrepancies between positives do not significantly violate the commonality semantics in general. Inspired by this, we implement the tolerable discrepancy to expand the original class boundary, and the boundary expansion process is depicted in Figure 3. Mathematically, the elements of the expanded boundary are transformed by the coordinates of the fuzzy representations, which are formalized as

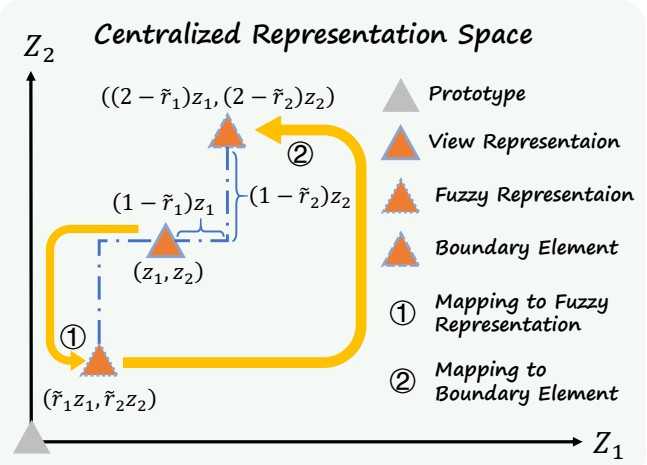

Figure 3: The boundary expansion process in 2-dimension space. One centralized view representation ▲ is firstly utilized to calculate the fuzzy representation ▲, and then mapped as the boundary element ▲.

$$\overline{\boldsymbol{Z}}^{(v)} = \widetilde{\boldsymbol{Z}}^{(v)} + 2\left(\boldsymbol{1} - \widetilde{\boldsymbol{R}}^{(v)}\right) \odot \boldsymbol{Z}^{(v)} = \left(\boldsymbol{2} - \widetilde{\boldsymbol{R}}^{(v)}\right) \odot \boldsymbol{Z}^{(v)}, \tag{8}$$

where $\overline{\boldsymbol{Z}}^{(v)}$ is composed of boundary elements, $\boldsymbol{1}$ and $\boldsymbol{2}$ are two real value matrices whose each element is 1 and 2 respectively. Though simple, the expanded boundary inherits two desired properties from the fuzzy representations: 1) **Semantics-positive**. The boundary elements are supposed to be the positives of crisp representations, which benefit recognizing the ambiguous samples far away from the class centers. 2) **Self-adaptive**. The expanded boundaries are based on the learnable membership functions and could adjust adaptively to involve the ambiguous samples in the training process. These two properties aim to capture nearly all the samples inside each boundary for fully exploiting the inter-class difference.

### 3.3.2 Loss Function

After expanding the original boundaries, we restore the coordinates of the centralized fuzzy representation matrix $\overline{Z}$ by adding their corresponding class prototype $C$ as the restored fuzzy representation matrix $\overline{H}$, since it is tough to separate the classes in a centralized representation space whose class centers are all origin points. The expansion of the original class boundaries leads to the new-born prototype of boundary elements $\widetilde{C}$, which causes a trade-off between $C$ and $\widetilde{C}$. We mix up both to harvest the fused prototype $\overline{C}$ by a linear combination as

$$\overline{C} = \eta C + (1 - \eta)\, \widetilde{C}, \tag{9}$$

where each prototype vector of $C$ is calculated by Eq. (3), $\widetilde{C}$ is calculated by averaging the boundary elements of each class, and $\eta$ is a balance coefficient to reconcile $C$ and $\widetilde{C}$.

To generalize for unseen data as much as possible, we expect to maximize the spectrum of each class, which is equal to maximizing the variance within each class. Meanwhile, we expect to achieve better separateness among different classes and compactness within each class, which intends to maximize the distances between different classes while minimizing the variance within each class. Considering these factors, the loss function is formalized as a *minimax game* by

$$\mathcal{L}_{pro} = \min_{\mathcal{P}+\mathcal{V}} \max_{\mathcal{V}} (\mathcal{P} + \mathcal{V}), \tag{10}$$

where $\mathcal{P}$ is defined to distance different classes and $\mathcal{V}$ to tighten samples within each class while generalizing for unseen data. They could be defined as

$$\begin{cases} \mathcal{P} = -\frac{1}{NKV} \sum_{v=1}^{V} \sum_{k=1}^{K} \sum_{n=1}^{N} \left\| \overline{h}_n^{(v)} - \overline{c}_k \right\|_2^2 \\ \mathcal{V} = \frac{1}{NV} \sum_{v=1}^{V} \sum_{k=1}^{K} \sum_{y_i=k} \left\| \overline{h}_i^{(v)} - \overline{c}_k \right\|_2^2 \end{cases} . \tag{11}$$

For convenience, we let $\mathcal{Q} = \mathcal{P} + \mathcal{V}$, and propose an upper bound of $\mathcal{Q}$ in our model, expecting to further optimize $\mathcal{Q}$ by optimizing its upper bound. We define the upper bound $\mathcal{Q}_{pro}$ as

$$\mathcal{Q}_{pro} = -\frac{1}{NV} \sum_{v=1}^{V} \sum_{k=1}^{K} \sum_{y_i=k} \log \frac{e^{-\tau_{pro} \left\| \overline{h}_i^{(v)} - \overline{c}_k \right\|_2^2}}{\sum_j e^{-\tau_{pro} \left\| \overline{h}_i^{(v)} - \overline{c}_j \right\|_2^2}}, \tag{12}$$

where $\overline{c}_k$ is the fused prototype vector of the $k$-th class, $\overline{h}_i^{(v)}$ is the restored boundary elements of the $i$-th instance in the $v$-th view, and $\tau_{pro}$ is the temperature coefficient to control the compactness of classes. Generally, higher $\tau_{pro}$ makes the classes more compact. We provide the theoretical proof that $\mathcal{Q}_{pro}$ is an upper bound of $\mathcal{Q}$ in the Appendix.

In the prototype-scale alignment loss, we integrate all the view representations to serve as the class samples since partial view information easily releases misleading information or loses critical information, which may lead to ambiguous decision boundaries. In this way, all the view information is gathered to collaboratively form the inter-class difference for delimiting the decision boundaries. Additionally, according to (Oord et al. (2018)), contrastive learning evaluates the mutual information between the raw view data and the restored boundary elements, namely

$$I\left( x_i, \left\{ \overline{h}_i^{(v)} \right\}_{v=1}^{V} \right) \geq \log\left( \mathcal{N} \right) - \mathcal{Q}_{pro}, \tag{13}$$

where $I\left( \cdot \right)$ is used to calculate the mutual information, $\mathcal{N}$ is the sample amount participated in the loss function. This inequation reveals that more negatives raise the lower bound and preserve more complete information from the raw data for better alignment performance. Typically, integrating all the view representations is a flexible way to multiply the amount of negatives.

### 3.3.3 Combination Operator Determination

The combination operator $Comb$ in Eq. (6) is crucial to the expanded boundaries. On the one hand, the $Comb$ operator cannot damage the self-adaptive ability of the expanded boundaries.

On the other hand, the $Comb$ operator controls the class variances to generalize for unseen data. The former is achieved by the learnable membership functions, and the latter is affected by the determination of the $Comb$.

**Theorem 1**. When employing $Min\ Pooling$ as the $Comb$ operator, the overall variance of each class is maximized. The proof is provided in the Appendix.

Therefore, when the ideal $Comb$ operator is defined as the minimal of the membership functions, the spectrum of each class is maximized. It means that the expanded boundary could generalize more latent (unseen) samples belonging to corresponding classes. Finally, the prototype-scale alignment loss $\mathcal{L}_{pro}$ in Eq. (10) is reduced as

$$\mathcal{L}_{pro} = \min -\frac{1}{NV}\sum_{v=1}^{V}\sum_{i=1}^{N}\log\frac{e^{-\tau_{pro}\left\|\overline{\boldsymbol{h}}_i^{(v)}-\overline{\boldsymbol{c}}_i\right\|_2^2}}{\sum_j e^{-\tau_{pro}\left\|\overline{\boldsymbol{h}}_i^{(v)}-\overline{\boldsymbol{c}}_j\right\|_2^2}}. \tag{14}$$

### 3.4 Classification Prediction

During the classification process, we employ a joint classifier that adopts the parameter sharing strategy, and it has several merits: 1) Shared parameters receive information from all the views simultaneously, which is conducive to constructing a classifier specific for the multi-view data. 2) The multi-scale alignment module has mined the inter-view commonality and the inter-class difference to construct clear decision boundaries, reducing the demand on the complex classifier with a large number of parameters. The model prediction is produced by averaging the outputs of the joint classifier by

$$\hat{\boldsymbol{p}}_i = \frac{1}{V}\sum_{v=1}^{V}\boldsymbol{p}_i^{(v)} = \frac{1}{V}\sum_{v=1}^{V}g\left(\boldsymbol{h}_i^{(v)}\right), \tag{15}$$

where $g\left(\cdot\right)$ is a logistic classifier, $\boldsymbol{p}_i^{(v)}$ is the output of $g\left(\cdot\right)$ in the $v$-th view, and $\hat{\boldsymbol{p}}_i$ is the final prediction. During training, we use the cross-entropy to calculate the classification loss as

$$\mathcal{L}_{cls} = -\frac{1}{N}\sum_{i=1}^{N}\sum_{j=1}^{K}\hat{\boldsymbol{y}}_{ij}\log\hat{\boldsymbol{p}}_{ij}. \tag{16}$$

where $\hat{\boldsymbol{p}}_i$ is the final prediction from Eq. (15), and $\hat{\boldsymbol{p}}_{ij}$ is its $j$-th element. $\hat{\boldsymbol{y}}_i$ refers to the one-hot vector label of the $i$-th instance, and $\hat{\boldsymbol{y}}_{ij}$ is the corresponding element in this vector. The overall loss $\mathcal{L}_{all}$ comprises four parts, namely reconstruction loss, classification loss, instance-scale alignment loss, and prototype-scale alignment loss, which is given by

$$\mathcal{L}_{all} = \mathcal{L}_{rec} + \mathcal{L}_{cls} + \alpha\mathcal{L}_{ins} + \beta\mathcal{L}_{pro}, \tag{17}$$

where $\alpha$ and $\beta$ are penalty coefficients. $\mathcal{L}_{all}$ focuses on mining the commonality among different views and clearing ambiguity areas around the decision boundaries, which successfully alleviates the damage of the feature heterogeneity and the information redundancy.

## 4 Experiments

In our experiment, we evaluate our model on eight public multi-view datasets including HandWritten, Scence15, PIE, CCV, Animal, 100Leaves, Hdigit, and YoutubeFace. To show the effectiveness of the proposed model, seven state-of-the-art methods are adopted to compare with, including **mm-dynamics** (Han et al. (2022a)), **ETMC** (Han et al. (2022b)), **UMDL** (Xu et al. (2023a)), **PDMF** (Xu et al. (2023b)), **IPMVSC** (Hu et al. (2023)), **MV-HFMD** (Black & Souvenir (2024)), and **RCML** (Xu et al. (2024)). For each dataset, we split 80% instances for training and the remainder for testing. To obtain reliable results, we implement the same dataset split ten times for all the models. More details of the datasets are provided in the Appendix.

### 4.1 Experimental Results

Table 1 presents the classification results between our model and other baseline models, and we adopt Accuracy (Acc), Purity, Recall, and Macro-F1 as the metrics. From the statistical comparisons, some important observations are revealed:

Table 1: Classification results (%) with baseline models. The best results are highlighted in bold. "OOM" indicates that the models raise the out-of-memory failure.

| Datasets | Metrics | mmdynamics | ETMC | UMDL | PDMF | IPMVSC | MV-HFMD | RCML | Ours |
|---|---|---|---|---|---|---|---|---|---|
| HandWritten | Acc | 98.5±0.5 | 96.8±0.7 | 97.7±0.5 | 98.2±0.7 | 98.7±0.4 | 98.2±0.6 | 97.1±0.9 | **98.9±0.4** |
| | Precision | 98.5±0.5 | 96.8±0.7 | 97.8±0.5 | 98.3±0.7 | 98.7±0.4 | 98.2±0.5 | 97.1±0.9 | **98.9±0.4** |
| | Recall | 98.5±0.5 | 96.9±0.7 | 97.7±0.5 | 98.3±0.7 | 98.7±0.4 | 98.2±0.6 | 97.2±0.9 | **98.9±0.4** |
| | Macro F1 | 98.5±0.5 | 96.8±0.7 | 97.7±0.5 | 98.2±0.7 | 98.7±0.4 | 98.1±0.5 | 97.1±0.9 | **98.9±0.4** |
| Scene15 | Acc | 62.0±2.2 | 66.5±1.8 | 63.3±0.4 | 67.8±1.1 | 71.8±1.4 | 80.7±1.2 | 70.0±1.0 | **81.5±0.7** |
| | Precision | 57.3±2.8 | 66.6±2.5 | 61.2±0.7 | 65.8±2.8 | 73.8±0.7 | 80.6±1.6 | 70.6±1.3 | **81.0±0.6** |
| | Recall | 60.1±2.2 | 65.3±1.6 | 62.7±0.5 | 65.9±1.1 | 70.5±0.8 | 80.3±1.3 | 68.9±1.3 | **80.8±0.7** |
| | Macro F1 | 57.6±2.5 | 62.8±1.8 | 61.1±0.7 | 63.0±1.4 | 69.1±0.9 | 80.0±1.4 | 67.1±1.1 | **80.5±0.7** |
| PIE | Acc | 72.5±3.7 | 90.5±2.3 | 71.1±3.3 | 88.7±3.0 | 91.9±2.0 | 86.8±2.4 | 91.8±2.9 | **93.1±1.7** |
| | Precision | 70.6±4.0 | 88.0±3.8 | 74.6±3.9 | 87.6±3.8 | 81.7±2.1 | 87.7±2.7 | 91.1±3.4 | **92.8±2.1** |
| | Recall | 70.6±3.7 | 89.6±3.6 | 71.1±3.3 | 87.3±4.9 | 82.1±1.8 | 88.9±2.4 | 91.0±3.5 | **93.6±1.8** |
| | Macro F1 | 67.4±3.8 | 87.4±3.8 | 69.3±3.5 | 85.7±4.5 | 80.5±2.5 | 83.8±6.1 | 89.8±3.9 | **92.1±2.0** |
| CCV | Acc | 29.6±1.1 | 42.6±1.4 | 36.2±1.5 | 50.5±1.4 | 42.6±1.8 | 53.4±0.9 | 42.4±1.7 | **54.0±1.1** |
| | Precision | 22.8±2.8 | 41.4±2.6 | 35.5±1.3 | 48.5±1.3 | 48.3±2.0 | 51.1±1.1 | 42.3±2.2 | **53.0±1.3** |
| | Recall | 23.2±1.0 | 37.1±1.4 | 33.7±1.4 | 45.4±1.3 | 35.5±1.3 | 50.5±0.8 | 37.0±1.4 | **50.6±1.0** |
| | Macro F1 | 20.1±1.1 | 36.0±1.6 | 33.1±1.4 | 45.1±1.5 | 34.3±1.5 | 50.4±0.9 | 35.9±1.5 | **51.0±0.9** |
| Animal | Acc | 56.7±1.4 | 56.6±0.8 | 34.0±1.1 | 57.0±0.7 | 39.5±1.9 | 59.6±0.7 | 56.8±1.2 | **60.2±1.1** |
| | Precision | 52.4±2.2 | 53.3±3.2 | 33.7±0.9 | 49.5±0.6 | 37.3±3.3 | 56.5±0.6 | 54.5±2.5 | **56.7±1.3** |
| | Recall | 49.9±1.3 | 49.7±1.0 | 30.4±0.8 | 47.2±0.7 | 33.3±2.5 | 53.7±0.4 | 49.5±1.2 | **53.8±0.8** |
| | Macro F1 | 50.4±1.4 | 49.7±1.1 | 30.4±0.7 | 46.7±0.6 | 33.3±2.7 | 54.0±0.5 | 49.6±1.3 | **54.1±0.8** |
| 100Leaves | Acc | 93.5±1.5 | 90.8±2.1 | 98.4±0.8 | 97.7±0.7 | 66.1±4.1 | 98.3±0.2 | 88.6±1.5 | **98.5±1.1** |
| | Precision | 93.6±1.2 | 90.5±2.5 | **98.7±0.8** | 97.7±0.9 | 70.2±3.4 | 98.4±0.0 | 88.7±2.0 | 98.5±1.0 |
| | Recall | 93.9±1.1 | 91.3±2.4 | 98.4±0.9 | 98.0±1.0 | 69.0±2.0 | 98.9±0.2 | 89.4±1.4 | **99.0±0.7** |
| | Macro F1 | 92.7±1.3 | 89.3±2.7 | 98.3±0.9 | 97.5±1.0 | 64.0±2.7 | 98.4±0.1 | 87.0±1.7 | **98.5±1.0** |
| Hdigit | Acc | 99.6±0.1 | 90.8±2.1 | 98.0±0.2 | 99.4±0.2 | 97.8±0.3 | 84.1±2.0 | 98.3±0.3 | **99.8±0.1** |
| | Precision | 99.6±0.1 | 98.4±0.2 | 98.0±0.2 | 99.4±0.2 | 97.8±0.3 | 88.9±1.7 | 98.3±0.3 | **99.8±0.1** |
| | Recall | 99.6±0.1 | 98.4±0.2 | 98.0±0.2 | 99.4±0.2 | 97.9±0.3 | 87.5±2.0 | 98.3±0.3 | **99.8±0.1** |
| | Macro F1 | 99.6±0.1 | 98.4±0.2 | 98.0±0.2 | 99.4±0.2 | 97.8±0.3 | 86.5±2.3 | 98.3±0.3 | **99.8±0.1** |
| YoutubeFace | Acc | 56.3±0.3 | 71.9±2.1 | OOM | 85.6±0.3 | 28.2±0.3 | 83.4±0.8 | 52.6±1.0 | **87.1±0.2** |
| | Precision | 74.8±0.6 | 83.2±0.8 | OOM | 89.2±0.4 | 29.9±4.0 | 85.6±1.0 | 83.1±1.0 | **89.5±0.4** |
| | Recall | 46.0±0.3 | 67.7±3.1 | OOM | 84.3±0.4 | 5.4±0.3 | 82.8±0.8 | 38.7±1.3 | **86.2±0.3** |
| | Macro F1 | 53.5±0.5 | 73.0±2.8 | OOM | 86.6±0.3 | 5.2±0.4 | 83.9±0.8 | 46.6±1.4 | **87.7±0.1** |

1) The proposed model shows extraordinary superiority among the comparing methods across all the datasets, especially on the Scene15 dataset. Compared with existing methods, our model is ideal to align all the view information while learning satisfactory decision boundaries, which are critical for the classification task.

2) For datasets that all the models receive wonderful results such as HandWritten, 100Leaves, and Hdigit datasets, the satisfactory decision boundaries are easily delimited. However, the proposed model could further improve the performance, which implies the rooted ambiguity near the decision boundaries is further alleviated.

3) MV-HFMD shows a very outstanding performance, which implies the significance of effective feature fusion in multi-view learning. However, it may neglect to address the feature heterogeneity that plays an important role in facilitating the feature fusion, so the model performance is somewhat limited when compared with our model.

4) Compared with a fuzzy multi-view learning model IPMVSC that focuses on using fuzzy rules to realize an interpretable classifier, our model attends to the feature-level issues and applies the fuzzy set theory in the representation space. Besides, from the results, we find that IPMVSC does not work well on large datasets, but our model works well on both small and large datasets.

In summary, the proposed model successfully mines the commonality to tackle the feature heterogeneity, learns the adaptive boundaries to mitigate the harm from the information redundancy, and preserves margins for unseen data to enhance the model generalization.

## 4.2 ABLATION STUDY

To evaluate the effectiveness of the multi-scale alignment module and the rationality of the $Comb$ operator determination, we conduct the ablation study whose results are provided in Table 2, and

Table 2: Results of ablation study on classification (%), which are concerned with the functions of the multi-scale alignment module and the selection of $Comb$ operator.

| $\mathcal{L}_{cls}$ | $\mathcal{L}_{rec}$ | $\mathcal{L}_{ins}$ | $\mathcal{L}_{pro}$ | $Comb$ | HandWritten | Animal | PIE | CCV |
|---|---|---|---|---|---|---|---|---|
| ✓ | ✓ | | | | 96.08±0.47 | 55.78±0.04 | 84.93±0.36 | 42.48±0.58 |
| ✓ | ✓ | ✓ | | | 97.44±0.54 | 56.95±0.42 | 89.71±1.20 | 49.34±3.33 |
| ✓ | ✓ | | ✓ | | 96.76±0.68 | 56.95±0.65 | 90.81±0.82 | 51.01±2.39 |
| ✓ | ✓ | ✓ | ✓ | | 97.46±1.10 | 57.80±0.15 | 91.54±0.82 | 51.69±2.18 |
| ✓ | ✓ | | ✓ | Min | 98.15±0.41 | 59.20±0.75 | 92.40±0.92 | 52.34±0.34 |
| ✓ | ✓ | ✓ | ✓ | Min | **98.85±0.42** | **60.16±1.03** | **93.11±1.72** | **53.97±1.14** |
| ✓ | ✓ | ✓ | ✓ | Max | 97.67±0.42 | 58.26±0.31 | 91.92±0.60 | 52.18±2.68 |
| ✓ | ✓ | ✓ | ✓ | Mean | 98.69±0.30 | 59.74±0.81 | 92.46±1.09 | 53.11±0.40 |

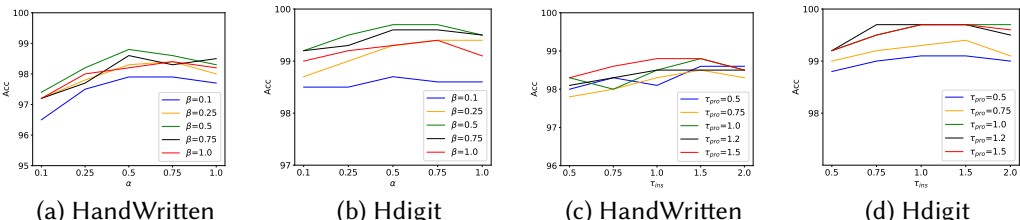

(a) HandWritten    (b) Hdigit    (c) HandWritten    (d) Hdigit

Figure 4: Hyperparameter analysis on the HandWritten and Hdigit datasets. (a) and (b) delve into the functions of penalty coefficients $\alpha$ and $\beta$, (c) and (d) delve into the functions of temperature coefficients $\tau_{ins}$ and $\tau_{pro}$.

have the following observations:

1) According to the results, the instance-scale alignment and the prototype-scale alignment work effectively to improve the inter-view commonalities and inter-class differences, which are essential for addressing the problems caused by feature heterogeneity and information redundancy.

2) In contrast to the instance-scale alignment, the prototype-scale alignment shows greater improvements, which is attributed to exploiting the helpful inter-class differences for delimiting the clear boundaries.

3) Compared with different $Comb$ operators, the $Min\ Pooling$ exhibits the best performance, which demonstrates the superiority of preserving more margins for classifying unseen data.

### 4.3 Hyperparameter Analysis

We conduct the hyperparameter analysis on two penalty coefficients $\alpha$ and $\beta$, and two temperature coefficients $\tau_{ins}$ and $\tau_{pro}$. For $\alpha$ and $\beta$, we vary their values from $\{0.1, 0.25, 0.5, 0.75, 1\}$, and for $\tau_{ins}$ and $\tau_{pro}$, we vary their values from $\{0.5, 0.75, 1.0, 1.5, 2.0\}$ and $\{0.5, 0.75, 1.0, 1.2, 1.5\}$ respectively. The results on HandWritten dataset and Hdigit dataset are presented in Figure 4. Compared with the temperature coefficients, the penalty coefficients affect the results more significantly, which illustrates the effectiveness of the multi-scale alignment module. Besides, the best results are prone to larger temperature coefficients that weigh more on similar samples.

### 5 Conclusion

Feature heterogeneity and information redundancy among the multi-view data are likely to cause the difficulty of feature fusion and the ambiguity around the decision boundaries, which motivates us to propose a novel MVL model MAMC for addressing the problems. MAMC employs the view-specific auto-encoders to abstract exclusive view representations and introduces a multi-scale alignment module to reduce the feature heterogeneity by mining the feature commonality. Besides, we propose a novel expanded boundary to exploit the class difference, which benefits to clear the ambiguity from the information redundancy. Extensive experiments demonstrate the superior performance and rational design of the proposed model.

ACKNOWLEDGMENTS

This work was supported by the National Natural Science Foundation of China (No. 62306020), the Young Elite Scientist Sponsorship Program by BAST (BYESS2024199), the National Key Research and Development Program of China (No. 2023YFB3107100), and the Beijing Natural Science Foundation (L244009). The authors gratefully acknowledge the support of K.C.WONG Education Foundation.

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

# A    ADDITIONAL PROOFS

## A.1    PROOF OF PROPOSITION 1

**Proposition 1**: $\mathcal{Q}_{pro}$ is an upper bound of $\mathcal{Q}$.

*Proof:* Without losing generalization, we let $\tau_{pro} = 1$, and we have

$$
\begin{aligned}
\mathcal{Q}_{pro} &= -\frac{1}{NV}\sum_{v=1}^{V}\sum_{k=1}^{K}\sum_{y_i=k}\log\frac{e^{-\tau_{pro}\left\|\overline{\boldsymbol{h}}_i^{(v)}-\overline{\boldsymbol{c}}_k\right\|_2^2}}{\sum_j e^{-\tau_{pro}\left\|\overline{\boldsymbol{h}}_i^{(v)}-\overline{\boldsymbol{c}}_j\right\|_2^2}} \\
&= \frac{1}{NV}\sum_{v=1}^{V}\sum_{k=1}^{K}\sum_{y_i=k}\log\frac{\sum_j e^{-\left\|\overline{\boldsymbol{h}}_i^{(v)}-\overline{\boldsymbol{c}}_j\right\|_2^2}}{e^{-\left\|\overline{\boldsymbol{h}}_i^{(v)}-\overline{\boldsymbol{c}}_k\right\|_2^2}} \\
&= \frac{1}{NV}\sum_{v=1}^{V}\sum_{k=1}^{K}\sum_{y_i=k}\left(\log\sum_j e^{-\left\|\overline{\boldsymbol{h}}_i^{(v)}-\overline{\boldsymbol{c}}_j\right\|_2^2} - \log e^{-\left\|\overline{\boldsymbol{h}}_i^{(v)}-\overline{\boldsymbol{c}}_k\right\|_2^2}\right) \\
&= \frac{1}{NV}\sum_{v=1}^{V}\sum_{k=1}^{K}\sum_{y_i=k}\left(\log\sum_j e^{-\left\|\overline{\boldsymbol{h}}_i^{(v)}-\overline{\boldsymbol{c}}_j\right\|_2^2} + \left\|\overline{\boldsymbol{h}}_i^{(v)}-\overline{\boldsymbol{c}}_k\right\|_2^2\right) \\
&\approx \frac{1}{NV}\sum_{v=1}^{V}\sum_{k=1}^{K}\sum_{y_i=k}\left(\log K\mathbb{E}\left[e^{-\left\|\overline{\boldsymbol{h}}_i^{(v)}-\overline{\boldsymbol{c}}_j\right\|_2^2}\right] + \left\|\overline{\boldsymbol{h}}_i^{(v)}-\overline{\boldsymbol{c}}_k\right\|_2^2\right) \\
&= \frac{1}{NV}\sum_{v=1}^{V}\sum_{k=1}^{K}\sum_{y_i=k}\left(\log\mathbb{E}\left[e^{-\left\|\overline{\boldsymbol{h}}_i^{(v)}-\overline{\boldsymbol{c}}_j\right\|_2^2}\right] + \left\|\overline{\boldsymbol{h}}_i^{(v)}-\overline{\boldsymbol{c}}_k\right\|_2^2 + \log K\right).
\end{aligned}
\tag{18}
$$

$K$ is a constant to denote the number of classes, and we ignore it to obtain a new loss $\mathcal{Q}'_{pro}$, namely

$$
\mathcal{Q}'_{pro} = \frac{1}{NV}\sum_{v=1}^{V}\sum_{k=1}^{K}\sum_{y_i=k}\left(\log\mathbb{E}\left[e^{-\left\|\overline{\boldsymbol{h}}_i^{(v)}-\overline{\boldsymbol{c}}_j\right\|_2^2}\right] + \left\|\overline{\boldsymbol{h}}_i^{(v)}-\overline{\boldsymbol{c}}_k\right\|_2^2\right).
\tag{19}
$$

According to the *Jensen's inequality*, we could get

$$
\exp\left\{\mathbb{E}\left[-\left\|\overline{\boldsymbol{h}}_i^{(v)}-\overline{\boldsymbol{c}}_j\right\|_2^2\right]\right\} \leq \mathbb{E}\left[\exp\left\{-\left\|\overline{\boldsymbol{h}}_i^{(v)}-\overline{\boldsymbol{c}}_j\right\|_2^2\right\}\right].
\tag{20}
$$

Considering that the logarithmic function is a monotonically increasing function, we have

$$
\mathcal{Q}'_{pro} \geq \frac{1}{NV}\sum_{v=1}^{V}\sum_{k=1}^{K}\sum_{y_i=k}\left(\mathbb{E}\left[-\left\|\overline{\boldsymbol{h}}_i^{(v)}-\overline{\boldsymbol{c}}_j\right\|_2^2\right] + \left\|\overline{\boldsymbol{h}}_i^{(v)}-\overline{\boldsymbol{c}}_k\right\|_2^2\right) \approx \mathcal{Q}.
\tag{21}
$$

## A.2    PROOF OF THEOREM 1

**Theorem 1**. When employing $Min\ Pooling$ as the $Comb$ operator, the overall variance of each class is maximized.

*Proof:* Without losing generalization, we let $\tau_{pro} = 1$, and we have

$$
\begin{aligned}
\mathcal{Q}_{pro} &= -\frac{1}{NV} \sum_{v=1}^{V} \sum_{k=1}^{K} \sum_{y_i=k} \log \frac{e^{-\left\|\overline{\boldsymbol{h}}_i^{(v)} - \overline{\boldsymbol{c}}_k\right\|_2^2}}{\sum_j e^{-\left\|\overline{\boldsymbol{h}}_i^{(v)} - \overline{\boldsymbol{c}}_j\right\|_2^2}} \\
&= \frac{1}{NV} \sum_{v=1}^{V} \sum_{k=1}^{K} \sum_{y_i=k} \left( \log \sum_j e^{-\left\|\overline{\boldsymbol{h}}_i^{(v)} - \overline{\boldsymbol{c}}_j\right\|_2^2} + \left\|\overline{\boldsymbol{h}}_i^{(v)} - \overline{\boldsymbol{c}}_k\right\|_2^2 \right) \\
&= \frac{1}{NV} \sum_{v=1}^{V} \sum_{k=1}^{K} \sum_{y_i=k} \left( \log \sum_j e^{-\left\|\overline{\boldsymbol{h}}_i^{(v)} - \overline{\boldsymbol{c}}_j\right\|^2} \right) + \frac{1}{NV} \sum_{v=1}^{V} \sum_{k=1}^{K} \sum_{y_i=k} \left( \left\|\overline{\boldsymbol{h}}_i^{(v)} - \overline{\boldsymbol{c}}_k\right\|_2^2 \right) \\
&\approx \frac{1}{NV} \sum_{v=1}^{V} \sum_{i=1}^{N} \left( \log \sum_j e^{-\left\|\overline{\boldsymbol{h}}_i^{(v)} - \overline{\boldsymbol{c}}_j\right\|_2^2} \right) + \sum_{k=1}^{K} Var(\overline{\boldsymbol{c}}_k),
\end{aligned}
\tag{22}
$$

where $Var\left(\overline{\boldsymbol{c}}_k\right)$ stands for the variance of the $k$-th class cluster. Actually, it could be observed that $\mathcal{V} = \frac{1}{NV} \sum_{v=1}^{V} \sum_{k=1}^{K} \sum_{y_i=k} \left\|\overline{\boldsymbol{h}}_i^{(v)} - \overline{\boldsymbol{c}}_k\right\|_2^2 = \sum_{k=1}^{K} Var(\overline{\boldsymbol{c}}_k)$. In practice, we weigh the crisp prototype more and set $\eta$ to approximate 1, then we have

$$
\lim_{\eta \to 1} \mathcal{V} = \lim_{\eta \to 1} \sum_{k=1}^{K} Var\left(\overline{\boldsymbol{c}}_k\right) = \sum_{k=1}^{K} Var\left(\boldsymbol{c}_k\right).
\tag{23}
$$

When applying the centralized representations for constructing the expanded boundary, the maximal overall variance within each class cluster is formalized as

$$
\begin{aligned}
\max \lim_{\eta \to 1} \mathcal{V} &= \max \sum_{k=1}^{K} Var(\boldsymbol{c}_k) = \max \sum_{v=1}^{V} \sum_{k=1}^{K} \sum_{y_i=k} \left\|\overline{\boldsymbol{h}}_i^{(v)} - \boldsymbol{c}_k\right\|_2^2 \\
&= \max \sum_{v=1}^{V} \sum_{k=1}^{K} \sum_{y_i=k} \left\|\overline{\boldsymbol{z}}_i^{(v)}\right\|_2^2 = \max \sum_{v=1}^{V} \sum_{k=1}^{K} \sum_{y_i=k} \left\|\left(\boldsymbol{2} - \tilde{\boldsymbol{r}}_i^{(v)}\right) \odot \boldsymbol{z}_i^{(v)}\right\|_2^2 \\
&= \max \sum_{v=1}^{V} \sum_{i=1}^{N} \left\|\left(\boldsymbol{2} - \tilde{\boldsymbol{r}}_i^{(v)}\right) \odot \boldsymbol{z}_i^{(v)}\right\|_2^2,
\end{aligned}
\tag{24}
$$

where $K$ is the number of classes. Obviously, for each representation, when each element of $\tilde{\boldsymbol{r}}_i^{(v)}$ becomes smaller, $\mathcal{V}$ becomes greater, which is in line with

$$
\left\|\left(\boldsymbol{2} - \left\{\tilde{\boldsymbol{r}}_i^{(v)}\right\}_{min}\right) \odot \boldsymbol{z}_i^{(v)}\right\|_2^2 \geq \left\|\left(\boldsymbol{2} - \left\{\tilde{\boldsymbol{r}}_i^{(v)}\right\}_{other}\right) \odot \boldsymbol{z}_i^{(v)}\right\|_2^2,
\tag{25}
$$

where $\left\{\tilde{\boldsymbol{r}}_i^{(v)}\right\}_{min}$, and $\left\{\tilde{\boldsymbol{r}}_i^{(v)}\right\}_{other}$ are the membership degree vectors, which are the outputs of *Min Pooling*, and other simple *Comb* operators such as *Mean Pooling*, or *Max Pooling*. Finally, we have

$$
\sum_{v=1}^{V} \sum_{i=1}^{N} \left\|\left(\boldsymbol{2} - \left\{\tilde{\boldsymbol{r}}_i^{(v)}\right\}_{min}\right) \odot \boldsymbol{z}_i^{(v)}\right\|_2^2 \geq \sum_{v=1}^{V} \sum_{i=1}^{N} \left\|\left(\boldsymbol{2} - \left\{\tilde{\boldsymbol{r}}_i^{(v)}\right\}_{other}\right) \odot \boldsymbol{z}_i^{(v)}\right\|_2^2.
\tag{26}
$$

This means that among the simple *Comb* operators, *Min Pooling* as the *Comb* could maximize the overall variance of each class.

## B  Experiment Settings

### B.1  Datasets

We provide the complete statistics of the used datasets in Table 3. HandWritten [1] and Hdigit [2] datasets are composed of handwritten digit images, PIE [3] and YoutubeFace [4] datasets collect the data on facial recognition and expression recognition, Scene15 [5] dataset contains images from 15 different scene categories, CCV [6] dataset comprises videos under complex conditions, Animal [7] dataset contains images of different animals, 100Leaves [8] dataset is composed of images from 100 different leaves.

Table 3: Statistics of datasets, which include the number of instances, classes, and the view dimensions.

| Dataset | Instances | Classes | Views | View dimensions |
|---|---|---|---|---|
| HandWritten | 2,000 | 10 | 6 | 240/76/216/47/64/6 |
| Scene15 | 4,485 | 15 | 3 | 20/59/40 |
| PIE | 680 | 68 | 3 | 484/256/279 |
| CCV | 6,773 | 20 | 3 | 20/20/20 |
| Animal | 11,673 | 20 | 4 | 2,689/2,000/2,001/2,000 |
| 100Leaves | 1,600 | 100 | 3 | 60/60/60 |
| Hdigit | 10,000 | 10 | 2 | 784/256 |
| YoutubeFace | 101,499 | 31 | 5 | 64/64/512/647/838 |

### B.2  Implementation Details

The proposed model is implemented in Pytorch and trained with an SGD optimizer. The dimension settings of the encoder and decoder are formed by $\{D_v, 1.4 \times 512, 1.2 \times 512, 512\}$ and $\{512, 0.6 \times D_v, 0.8 \times D_v, D_v\}$ respectively. The number of membership functions $L$ is set as 5. The learning rate is chosen from $\{1e^{-3}, 3e^{-3}, 5e^{-3}, 1e^{-2}\}$, and the coefficients $\alpha$ and $\beta$ are chosen from $\{0.1, 0.25, 0.5, 0.75, 1\}$. The temperature coefficients $\tau_{ins}$ and $\tau_{pro}$ are chosen from $\{0.5, 0.75, 1.0, 1.5, 2.0\}$ and $\{0.5, 0.75, 1.0, 1.2, 1.5\}$ respectively. All experiments are conducted on a server with 8 NVIDIA GeForce 3090 (24 GB memory each).

### B.3  Training Algorithm

The proposed MAMC includes three main components of the proposed model, namely view-specific representation extractors, a multi-scale alignment module, and a joint classifier. Among the three components, the view-specific auto-encoders abstract critical view representations from multi-view data, and the multi-scale alignment module mines the inter-view commonality and inter-class difference. The joint classifier calculates the fused predictions. Algorithm 1 presents the detailed training process of the proposed MAMC.

## C  Supplementary Experiments

### C.1  Learning Clear Decision Boundaries

Information redundancy in the multi-view data would lead to ambiguous boundaries. To further verify that MAMC has the ability to learn clear decision boundaries, we implement our model on

---

[1] HandWritten: http://archive.ics.uci.edu/ml/datasets/Multiple+Features.

[2] Hdigit: https://archive.ics.uci.edu/ml/index.php.

[3] PIE: https://www.cs.cmu.edu/afs/cs/project/PIE.

[4] YoutubeFace: https://www.cs.tau.ac.il/wolf/ytfaces.

[5] Scene15: http://www-cvr.ai.uiuc.edu/ponce_grp/data/scene15.

[6] CCV: http://www.ee.columbia.edu/ln/dvmm/CCV.

[7] Animal: https://www.cs.ucf.edu/ xmzhang/datasets.

[8] 100Leaves: https://archive.ics.uci.edu/ml/datasets/One-hundred+plant+species+leaves+data+set.

---

**Algorithm 1** Training process of MAMC

---

**Input:** Multi-view dataset: $\mathcal{D} = \{(\{\boldsymbol{x}_i^{(v)}\}_{v=1}^V, y_i)|1 \leq i \leq N\}$; Hyperparameters $\tau_{ins}$, $\tau_{pro}$, $\alpha$, and $\beta$; Training epochs $T$.
**Output:** Model parameters.
**Initialization:** Initialize the parameters of the neural network.
**Process:**
 1: **for** $epoch = 1$ **to** $T$ **do**
 2:     Calculate the view representation matrices $\{\boldsymbol{H}^{(v)}\}_{v=1}^V$ with view-specific auto-encoders.
 3:     \\ Instance-scale alignment
 4:     Realize the instance-scale alignment by Eq. (2).
 5:     \\ Prototype-scale alignment
 6:     Centralize the view representations by Eq. (3)-(4).
 7:     Calculate the boundary elements by Eq. (5)-(8).
 8:     Realize the prototype-scale alignment by Eq. (14).
 9:     \\ Classification prediction
10:     Calculate the classification loss by Eq. (15)-(16).
11: **end for**

---

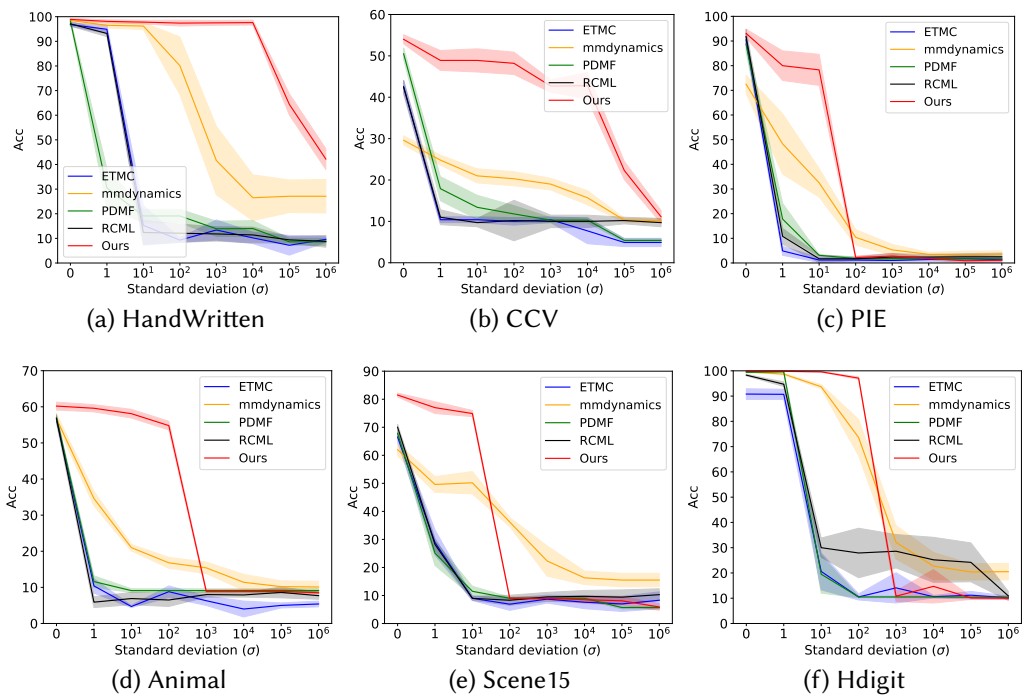

Figure 5: Performance comparison on multi-view datasets with different levels of noise.

manufactured ambiguous data. In our experiments, we add Gaussian noise with different levels of deviations $\sigma$ to half of the views, and provide the comparison results in Figure 5. We discover that when the noise increases, the performance of all the models drops quickly since larger noise results in more ambiguous boundaries. Even so, the performance of our model descends more slowly than other models, especially when the noise is small. Actually, the data noise in the real multi-view data is not large, which demonstrates that MAMC exploits the desired inter-difference from all the view information to clear the ambiguous decision boundaries.

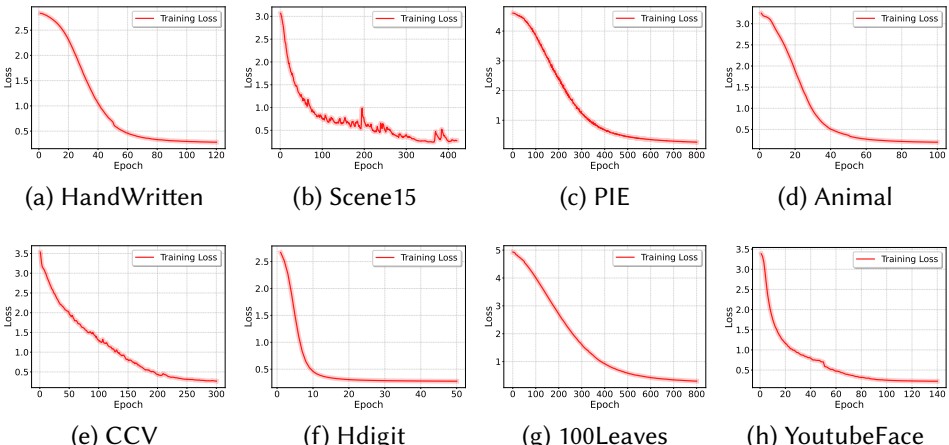

Figure 6: The convergence analysis of MAMC across all the datasets.

## C.2 Loss Analysis

We conduct the convergence analysis of the proposed method across all the datasets, where experimental results of loss values are illustrated in Figure 6. It is observed that the losses decrease steadily and converge between 50 and 800 epochs across all datasets. Such a phenomenon empirically confirms the convergence of MAMC.

## C.3 Visualization

In this experiment, we provide the t-SNE of the comparative MVL models and raw data in Figure 7. In practice, one instance is encoded as several view representations, which are then concatenated as a final representation vector. The final representation vector is input into the t-SNE algorithm to obtain the visualization. From the results, we could find that though the raw data are ambiguous, most models disentangle different classes to some degree. However, our model shows much better results since it mines the feature commonalities to tighten samples inside each class and exploits inter-class differences for separating different classes.

## C.4 Complexity Comparison

We present the model complexity comparison in Table 4 and Table 5. Table 4 lists the complexity of all the models, and we can find that our model has the same level of complexity as some comparative models. Table 5 provides detailed computational overhead analysis including MACs, concrete running time, and Parameters. For a fair comparison, we set the same batch size and training epochs for all the models. From all the statistics, we could find that our model is not cumbersome, but is competitive with some advanced models.

Table 4: Complexity comparison between the comparative models and the proposed model.

| Model | Complexity |
|---|---|
| mmdynamics | $\mathcal{O}(N)$ |
| ETMC | $\mathcal{O}(N)$ |
| UMDL | $\mathcal{O}(N^3)$ |
| PDMF | $\mathcal{O}(N^2)$ |
| RCML | $\mathcal{O}(N)$ |
| Ours | $\mathcal{O}(N^2)$ |

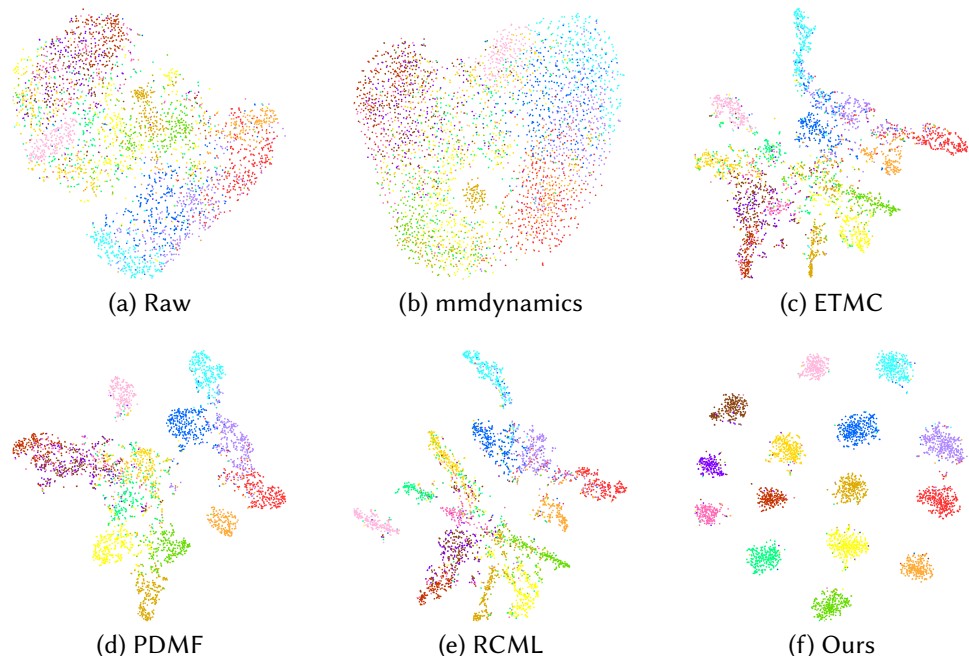

Figure 7: t-SNE of the comparative models and the proposed model on Scene15 dataset.

Table 5: Complexity comparison among the comparative models and the proposed model.

| Datasets | Metrics | mmdynamics | ETMC | UMDL | PDMF | RCML | Ours |
|---|---|---|---|---|---|---|---|
| HandWritten | Running Time | 21.911s | 35.739s | 10057.537s | 21.972s | 17.064s | 23.302s |
| | MACs | 267.871M | 3.323M | 1273.548M | 189.989M | 1.661M | 1373.655M |
| | Parameters | 1.0543M | 0.0066M | 5.2389M | 1.722M | 0.0066M | 5.353M |
| Scene15 | Running Time | 28.996s | 42.596s | 31938.726s | 45.678s | 22.922s | 41.651s |
| | MACs | 66.604M | 0.457M | 132.890M | 68.387M | 0.457M | 617.869M |
| | Parameters | 0.264M | 0.002M | 20.632M | 1.098M | 0.002M | 3.197M |
| PIE | Running Time | 11.525s | 11.233s | 2721.368s | 9.214s | 7.775s | 17.499s |
| | MACs | 536.394M | 17.739M | 248.090M | 188.088M | 2.217M | 996.040M |
| | Parameters | 2.099M | 0.070M | 1.429M | 1.540M | 0.070M | 4.622M |
| CCV | Running Time | 40.678s | 61.986s | 29783.373s | 69.529s | 37.905s | 65.490s |
| | MACs | 56.521M | 0.307M | 125.338M | 9.978M | 0.307M | 263.633M |
| | Parameters | 0.225M | 0.001M | 46.361M | 1.331M | 0.001M | 3.323M |
| Animal | Running Time | 650.847s | 181.911s | 52103.473s | 221.021s | 119.074s | 177.946s |
| | MACs | 7643.710M | 44.493M | 1269.197M | 576.355M | 44.493M | 9360.186M |
| | Parameters | 29.873M | 0.174M | 141.214M | 10.937M | 0.174M | 37.353M |
| 100Leaves | Running Time | 12.697s | 16.173s | 7386.190s | 19.825s | 10.078s | 44.393s |
| | MACs | 247.370M | 4.915M | 142.234M | 80.740M | 4.915M | 672.518M |
| | Parameters | 0.971M | 0.020M | 3.113M | 0.814M | 0.020M | 3.323M |
| Hdigit | Running Time | 57.370s | 58.236s | 44589.832s | 85.163s | 41.918s | 59.256s |
| | MACs | 506.520M | 0.666M | 211.558M | 42.877M | 2.662M | 883.593M |
| | Parameters | 1.982M | 0.010M | 100.825M | 2.614M | 0.010M | 4.245M |
| YoutubeFace | Running Time | 1660.614s | 598.449s | — | 513.350s | 523.801s | 702.112s |
| | MACs | 4393.923M | 67.456M | — | 1466.827M | 67.456M | 7988.087M |
| | Parameters | 4.328M | 0.066M | — | 15.840M | 0.066M | 8.544M |

