# OpenReview forum: "Enhance Multi-View Classification Through Multi-Scale Alignment and Expanded Boundary"
_ICLR.cc/2025/Conference — ICLR 2025 Poster_

### Official Review · Reviewer_8qZm · 2024-10-29

**Soundness:** 4
**Presentation:** 3
**Contribution:** 4
**Rating:** 8
**Confidence:** 5

**Summary:**

This paper delves into the problems in multi-view data, namely fusion difficulty caused by feature heterogeneity and boundary ambiguity caused by information redundancy. To tackle these problems, this paper proposes a novel multi-view learning model MAMC, which reduces the fusion difficulty by introducing a multi-scale alignment strategy to mine the inter-view commonality and inter-class difference, while alleviating the boundary ambiguity by proposing an expanded boundary to learn a clear decision boundary. Overall, this model is interesting and works well to address the problems.

**Strengths:**

- The paper structure is logical and the motivation is clear and easy to understand.

- The design of the expanded boundary is detailed, actually, it is interesting and seems simple-yet-effective.

- The authors provide the theoretical proof to explain the construction of the expanded boundary.

- The MAMC has very good performance in the classification results and visualization.

**Weaknesses:**

- It would be better for the authors to provide the additional results with the general prototype in the ablation study, which makes it easier to understand the advantage of the expanded boundary.

- The model is simple-yet-effective, however, I don’t know whether it is efficient. So the authors could provide a complexity analysis to understand the model efficiency better.

**Questions:**

- Though the visualization is quite good, I am confused about how they obtained the results. They should explain how to combine the multi-view features to realize t-SNE visualization, such as concatenation or other ways.

- I find the comparative models in the paper mostly deal with uncertainty. Thus, I want to know whether the proposed model can be regarded as one robust model to handle uncertainty. And what are the differences between those models and MAMC?

- Whether the authors give the wrong symbols in Figure 4? As the caption described, the symbols in (c) and (d) should be two temperature coefficients, and the authors need to check it up.

---

> ### Author Response · Authors · 2024-11-18
> **Our Response to the Weaknesses and Questions**
>
> **W: Weakness; Q: Question**
>
> **W1.** We supplement results with the general prototypes on four datasets in the ablation study, which are shown in Table 1. For convenience, we also provide the results of the model with the expanded boundary. From the results, we can find that the model using the general prototype loss exhibits inferior performance compared with the models using the expanded prototypes, which means the expanded boundary could further improve the prototype-scale alignment module.
>
> Table 1: Supplementary ablation analyses on multi-scale module.
> |$\mathcal{L}_{cls}$|$\mathcal{L}_{rec}$|$\mathcal{L}_{ins}$|$\mathcal{L}_{pro}$|Comb|HandWritten|Animal|PIE|CCV|
> |-|-|-|-|-|-|-|-|-|
> |√|√||||96.08±0.47|55.78±0.04|84.93±0.36|42.48±0.58|
> |√|√|√|||97.44±0.54|56.95±0.42|89.71±1.20|49.34±3.33|
> |√|√| |√||96.76±0.68|56.95±0.65|90.81±0.82|51.01±2.39|
> |√|√|√|√||97.46±1.10|57.80±0.15|91.54±0.82|51.69±2.18|
> |√|√|√|√|Min|98.85±0.42|60.16±1.03|93.11±1.72|53.97±1.14|
>
> **W2.** We provide the elaborate complexity analysis of MAMC, which comprises representation extractors, multi-scale alignment module, and joint classifier three parts. The representation extractors need $\mathcal{O}(NVd)$ complexity, the instance-scale alignment needs $\mathcal{O}(N^2V^2d)$ complexity, the prototype-scale alignment needs $\mathcal{O}(4NVd+NVKd)$ complexity, the joint classifier needs $\mathcal{O}(NVd)$ complexity. Then, the overall complexity is $\mathcal{O}(N^2V^2+6NVd+NVKd)$.
>
> **Q1.** The t-SNE visualization is obtained by concatenating the representation vectors from all the views. Specifically, one instance would be encoded as several view representations, which are then concatenated as a final representation vector. The final representation vector is input into the t-SNE algorithm to obtain the visualization. To provide better visualization of the comparative models, we apply the hyperparameter settings provided in their official codes, which would present satisfactory results.
>
> **Q2.**  As you mentioned, MAMC can be regarded as a robust model to handle uncertainty by integrating fuzzy set theory, which quantifies how much the samples belong to a class. And the differences between the comparative models and MAMC lie in the intentions of the models and the tools to deal with the uncertainties.
>
> Firstly, the intentions of the comparative models and MAMC are different. For the comparative models, they delve into improving the inferior logits in the decision phase but MAMC focuses on learning clear boundaries in the representation space. In our opinion, the data uncertainties will hurt the representation learning and decision phases, which implies that modifying the logits might be sub-optimal. Therefore, we focus on learning clear decision boundaries in the representation learning phase, which may handle the data uncertainties more directly.
>
> Additionally, with respect to the tools, comparative models are based on Dempster-Shafer Theory of Evidence to quantify uncertainty, which is a generalization of the Bayesian theory to subjective probabilities [1]. MAMC combines with fuzzy set theory which adopts the learnable membership function to represent varying degrees of uncertainty [2]. In summary, the comparative models focus on learning ideal subjective probabilities while MAMC aims at learning appropriate membership functions.
>
> [1] Evidential deep learning to quantify classification uncertainty. NeurIPS, 2018.
>
> [2] Robust TSK fuzzy system based on semisupervised learning for label noise data. IEEE TFS, 2020.
>
> **Q3.**  Thanks for your check, we will revise the wrong symbols in the final version.

---

> > ### Comment · Reviewer_8qZm · 2024-11-19
> > **see comment**
> >
> > Thank the authors' efforts, my concerns have been addressed. Moreover, I have read the comments from other reivewers and found that the reviewers approve that the technique is sound or method is novel. Indeed, there are some issues like missing important experiments, loss term setting, missing related work, unclear figure in the previous version. The authors have well responded them. Based on above analysis, I think the proposed method is very helpful for the trusted multi-modal learning  community. So I am willing to incease my score.

---

> > > ### Author Response · Authors · 2024-11-19
> > > **Thanks for your recommendation**
> > >
> > > Thank you very much for your recommendation to accept our paper. We deeply appreciate your valuable comments and insightful suggestions throughout the review process. Your feedback has significantly contributed to the improvement of our work, and we are truly grateful for your time and effort.

---

### Official Review · Reviewer_9Q5P · 2024-11-01

**Soundness:** 2
**Presentation:** 3
**Contribution:** 2
**Rating:** 5
**Confidence:** 4

**Summary:**

This papers addresses two issues in multi-view data: feature heterogeneity and information redundancy, by proposing a multi-view representation learning approach that enhances multi-view classification through multi-scale alignment and boundary expansion. Specifically, to address feature heterogeneity, this paper proposes a multi-scale alignment module that aligns from both instance and prototype scales, to explore commonalities from both inter-view and inter-class perspectives, thereby mitigating the issue of heterogeneity. To address information redundancy, this paper proposes a new expanded boundary method that adaptively adjusts boundaries using fuzzy set theory to handle fuzzy data. By integrating this expanded boundary with the prototype scale alignment module, the representation can be tightened and boundary ambiguity reduced, thereby enhancing the model's generalization capabilities. Experiments show that this model outperforms existing methods on multiple datasets.

**Strengths:**

This article utilizes fuzzy sets to propose a novel self-adaptive expanded boundary method to enhance the separability of inter-class representations and the compactness of intra-class representations in multi-view data.

**Weaknesses:**

1. The multi-scale alignment module proposed in this paper is an incremental approach. It is a commonly used paradigm in existing multi-view representation learning methods to enhance the discriminability and semantic consistency of representations.
2. The comparative methods in the experimental section are not comprehensive or representative enough. A thorough investigation of related work in the past three years should be conducted to demonstrate the effectiveness of this method.
3.One of the main problems addressed in this paper is feature heterogeneity, which is explained as the fusion difficulty caused by heterogeneity among multiple different data distributions. However, the proposed multi-scale alignment module does not provide an in-depth analysis of this issue or explain how it solves the problem.
4. The explanation of symbols in the paper is not clear enough, such as in Eq.(16).
5. The experimental section lacks an analysis of model convergence.
6.The datasets have not been cited.

**Questions:**

Please refer to Weaknesses.

---

> ### Author Response · Authors · 2024-11-18
> **Response to Weakness #1**
>
> **W1.** The multi-scale alignment module in this paper is the carrier for the expanded boundary to enhance the ability to tackle feature heterogeneity. Though the multi-scale alignment module is commonly used to improve discriminability and semantic consistency of representations, existing works rarely focus on further enhancing these abilities through learning effective boundaries. To address this gap, we propose the novel expanded boundary as one embedded component of the multi-scale alignment module to reinforce the discriminability and semantics consistency while clearing the ambiguous boundaries. Additionally, we provide theoretical guidance to ensure the alignment loss is both reasonable and reliable. Ablation studies demonstrate that our approach could further improve discriminability and semantic consistency by showing better model performance.

---

> ### Author Response · Authors · 2024-11-18
> **Response to the first question of Weakness #2**
>
> **W2.** We have selected a range of advanced comparative methods proposed in the past three years for our analysis. Among these, ETMC [1] is a well-known multi-view classification method that has been widely referenced and further developed by subsequent studies [2-3]. Besides, most comparative models focus on realizing trustworthy classification and could address data uncertainties which is also one merit of our model. And we have conducted the relevant experiments to handle noisy data in Appendix C.1. Therefore, these comparative methods are appropriate and representative.
> To offer a more comprehensive comparison, we also conducted additional experiments using some recent multi-view classification methods [4-5], with results presented in Table 1. Furthermore, we supplement the related works in the past three years and discuss the relevance between the comparative models and the proposed method below to demonstrate the effectiveness of our method, which will be added in the final version of the paper.
>
> **Investigation of related work:**
> Existing multi-view classification methods can generally be divided into two categories: feature fusion-based and decision-based methods. Feature fusion-based methods focus on effectively combining features from all views. MV-HFMD [4] explores feature fusion by introducing a novel fusion scheme and mutual distillation to adapt neural networks for multi-view classification. Mmdynamics [6] dynamically assesses the informativeness of both feature-level and view-level data across different samples, ensuring trustworthy integration of multiple views. However, these methods often neglect feature heterogeneity, which undermines the effectiveness of feature fusion. For the decision-based methods, they are dedicated to making more reasonable decisions. ETMC [1] introduces a new paradigm for multi-view learning by dynamically integrating different views at the evidence level, thereby enhancing classification reliability. RCML [2] attends to that conflictive information in different views, and proposes a conflictive opinion aggregation strategy that exactly models the relation of multi-view common and view-specific reliabilities. MVAKNN [3] utilizes the Dempster–Shafer theory to integrate information from each view, successfully extending adaptive KNN to a multi-view setting. IPMVSC [5] constructs a multi-view fuzzy classification model inherited from the natural interpretability of fuzzy rule-based systems to realize interpretable classification. Many methods of this kind interact with the multiple views only in the decision phase, which cannot well integrate the information from all the views. In contrast, our model aligns view features during the representation learning phase and combines them in the decision phase, allowing for more effective and interactive information integration across all views.
>
> Table 1: Classification results for supplemented comparative models.
> |Datasets|Metrics|MV-HFMD[3]|IPMVSC[4]|
> |-|-|-|-|
> |HandWritten|Acc|98.15±0.58|98.70±0.41|
> ||Precision|98.18±0.51|98.69±0.43|
> ||Recall|98.15±0.56|98.72±0.39|
> ||Macro F1|98.14±0.53|98.68±0.41|
> |Scene15|Acc|80.65±1.19|71.84±1.41|
> ||Precision|80.58±1.57|73.82±0.66|
> ||Recall|80.33±1.28|70.51±0.82|
> ||Macro F1|80.00±1.44|69.13±0.87|
> |PIE|Acc|86.76±2.41|91.91±2.03|
> ||Precision|87.65±2.74|81.72±2.14|
> ||Recall|88.86±2.41|82.11±1.84|
> ||Macro F1|83.75±6.11|80.54±2.54|
> |CCV|Acc|53.36±0.86|42.59±1.83|
> ||Precision|51.08±1.12|48.32±1.99|
> ||Recall|50.53±0.83|35.47±1.27|
> ||Macro F1|50.39±0.89|34.30±1.51|
> |Animal|Acc|59.58±0.67|39.51±1.93|
> ||Precision|56.47±0.55|37.34±3.88|
> ||Recall|53.68±0.40|33.31±2.46|
> ||Macro F1|54.01±0.46|33.33±2.72|
> |100Leaves|Acc|98.28±0.16|66.06±4.14|
> ||Precision|98.36±0.01|70.19±3.42|
> ||Recall|98.93±0.24|68.97±2.03|
> ||Macro F1|98.40±0.10|63.95±2.67|
> |Hdigit|Acc|87.40±2.0|97.84±0.28|
> ||Precision|88.90±1.67|97.83±0.28|
> ||Recall|87.52±1.99|97.86±0.26|
> ||Macro F1|86.49±2.26|97.83±0.27|
> |YoutubeFace|Acc|83.42±0.77|28.18±0.32|
> ||Precision|85.62±1.07|29.92±3.97|
> ||Recall|82.83±0.81|5.43±0.28|
> ||Macro F1|83.94±0.75|5.15±0.35|
>
>
> [1] Trusted multi-view classification with dynamic evidential fusion. IEEE TPAMI, 2023.
>
> [2] Reliable conflictive multi-view learning. AAAI 2024.
>
> [3] Multi-view adaptive k-nearest neighbor classification. IEEE TAI 2023.
>
> [4] Multi-view Classification Using Hybrid Fusion and Mutual Distillation. WACV 2024.
>
> [5] Multi-view fuzzy classification with subspace clustering and information granules. IEEE TKDE, 2023.
>
> [6] Multimodal dynamics: Dynamical fusion for trustworthy multimodal classification. CVPR 2022.

---

> ### Author Response · Authors · 2024-11-18
> **Response to the second question in Weakness #2**
>
> **W2-2.** Feature fusion could be divided into several phases, such as early fusion and late fusion. Our model adopts late fusion, with a focus on addressing feature heterogeneity to facilitate more effective fusion. To tackle the feature heterogeneity arising from different data distributions, our model is designed to minimize the distances between similar samples across different views. By utilizing multi-scale alignment modules, similar samples are gathered together, reducing the distances between them and aligning the data distributions across views. Furthermore, the embedded expanded boundary adaptively gathers more similar samples, further minimizing distribution differences. By reducing these distribution discrepancies across views, the representations can be more easily fused in the late fusion phase.
> To provide a clearer and more comprehensive analysis, we present the results of an ablation study in Table 2. The findings indicate that the multi-scale alignment module could enhance the quality of the fused representations, effectively mitigating feature heterogeneity.
>
> Table 2: Ablation analysis on the multi-scale module.
> |$\mathcal{L}_{cls}$|$\mathcal{L}_{rec}$|$\mathcal{L}_{ins}$|$\mathcal{L}_{pro}$|Comb|HandWritten|Animal|PIE|CCV|
> |-|-|-|-|-|-|-|-|-|
> |√|√||||96.08±0.47|55.78±0.04|84.93±0.36|42.48±0.58|
> |√|√|√|||97.44±0.54|56.95±0.42|89.71±1.20|49.34±3.33|
> |√|√| |√||96.76±0.68|56.95±0.65|90.81±0.82|51.01±2.39|
> |√|√|√|√||97.46±1.10|57.80±0.15|91.54±0.82|51.69±2.18|
> |√|√|√|√|Min|98.85±0.42|60.16±1.03|93.11±1.72|53.97±1.14|

---

> ### Author Response · Authors · 2024-11-18
> **Response to Weaknesses #3**
>
> **W3.**
>
> $N$ is the number of instances,
>
> $K$ is the number of classes,
>
> ${\hat{p}}_i$ is the final prediction from Eq. (15),
>
> ${\hat{p}}_{ij}$ is the $j$-th element of ${\hat{p}}_i$,
>
> ${\hat{y}}_i$ refers to the one-hot vector label of the $i$-th instance,
>
> and ${\hat{y}}_{ij}$ is the $j$-th element of ${\hat{y}}_i$.

---

> ### Author Response · Authors · 2024-11-18
> **Response to Weakness #4**
>
> **W4-1.** In our results, the losses decrease steadily and converge between 50 and 800 epochs across all datasets. However, as we are unable to provide the loss convergence figures in the rebuttal, we will include a detailed convergence analysis in the final version of the paper.
>
> **W4-2.** Thanks for your advice. We will add the related references/websites of our employed datasets in the final version of our paper, which are now listed as follows:
>
> (a) HandWritten: http://archive.ics.uci.edu/ml/datasets/Multiple+Features
>
> (b) PIE: https://www.cs.cmu.edu/afs/cs/project/PIE/
>
> (c) YoutubeFace: https://www.cs.tau.ac.il/wolf/ytfaces/
>
> (d) Animal: https://www.cs.ucf.edu/~xmzhang/datasets/
>
> (e) CCV: http://www.ee.columbia.edu/ln/dvmm/CCV/
>
> (f) 100Leaves: https://archive.ics.uci.edu/ml/datasets/One-hundred+plant+species+leaves+data+set
>
> (g) Hdigit: https://archive.ics.uci.edu/ml/index.php
>
> (h) Scene15: http://www-cvr.ai.uiuc.edu/ponce_grp/data/scene15/

---

> ### Author Response · Authors · 2024-11-21
> **Response**
>
> Dear Reviewer 9Q5P，
>
> Thank you for your time. We would like to kindly remind you that we have responded to your valuable feedback on our submission . If you have any further questions, comments, or concerns, we would be more than happy to address them.
>
> We deeply appreciate the time and effort you have dedicated to reviewing our work and providing constructive feedback, which has been immensely helpful to us.
>
> Please feel free to reach out if there is anything else we can clarify. Thank you again for your contribution to improving our research.
>
> Best regards,
>
> Authors.

---

> > ### Comment · Reviewer_9Q5P · 2024-11-26
> >
> > Thank you for the clarifications and additional experimental results. The responses addressed my concerns, and I will keep my score.

---

> > > ### Author Response · Authors · 2024-11-26
> > >
> > > Thank you very much for your thoughtful comments and for confirming that our responses have addressed your concerns. We deeply appreciate the time and effort you have dedicated to reviewing our work.
> > >
> > > We noticed that your current score remains on the borderline below acceptance. While we respect your evaluation, we would be grateful if you could let us know if there are any other aspects of the paper—such as the contribution, motivation, or novelty—that you feel require further clarification or improvement. Your feedback would be invaluable in helping us better communicate the merits of our work.
> > >
> > > Once again, thank you for your constructive feedback and for contributing to the improvement of our paper.
> > >
> > > Best regards,
> > >
> > > Authors.

---

### Official Review · Reviewer_1xbb · 2024-11-03

**Soundness:** 3
**Presentation:** 2
**Contribution:** 2
**Rating:** 5
**Confidence:** 4

**Summary:**

This paper proposes a multi-scale alignment module and an expanded boundary based on fuzzy set theory to address feature heterogeneity and information redundancy in multi-view classification. The alignment module enhances feature fusion, while the expanded boundary adapts to ambiguous data, reducing boundary ambiguity and improving generalization.

**Strengths:**

1. The proposed MAMC model effectively addresses feature heterogeneity by employing view-specific auto-encoders and a multi-scale alignment module, which extracts common features across views, enhancing feature fusion and improving classification performance.
2. The model introduces an expanded boundary to manage information redundancy and class ambiguity, leveraging class differences to create clearer, more adaptive decision boundaries, which aids in distinguishing classes more effectively.

**Weaknesses:**

1. The details of the paper need further refinement. For example, in subfigures (c) and (d) of Figure 4, the axis labels should use "temperature coefficients" instead of "penalty coefficients."
2. Equation 17 represents the overall optimization objective, involving four different loss functions. Why do the penalty coefficients apply to only two of these loss functions?
3. Figure 5 in the appendix should include variance, as it is a key aspect of stability analysis.
4. There are many minor issues that need refinement. For example, the first column in Table 1 should be labeled "datasets" instead of "methods."
5. The experiments were conducted using eight 24GB 3090 GPUs, but there are still out-of-memory issues reported in Table 1. This clearly indicates that the model's complexity and computational overhead require analysis and comparison, including metrics such as runtime, MACs, FLOPs, and parameters.

**Questions:**

Please see weakness.

---

> ### Author Response · Authors · 2024-11-18
> **Response to Weakness #1-- #4**
>
> **W1.** Thanks for your check, we will revise the wrong symbols in the final version of the paper.
>
> **W2.** The construction loss and the cross-entropy loss are two common losses in multi-view learning [1-3]. And we follow some works [2], [4] to fix the penalty coefficients of these terms to be one. By fixing the coefficients, we can get deep insights into the other two losses that correspond to the proposed modules. Meanwhile, the complexity of hyperparameter search will also be mitigated to some degree.
>
> [1] Deep double incomplete multi-view multi-label learning with incomplete labels and missing views. IEEE TNNLS 2023.
>
> [2] Triple-granularity contrastive learning for deep multi-view subspace clustering. ACM MM 2023.
>
> [3] Self-supervised information bottleneck for deep multi-view subspace clustering. IEEE TIP 2023.
>
> [4] MMatch: Semi-supervised discriminative representation learning for multi-view classification. IEEE TCSVT 2022.
>
> **W3.** Because of the inadmissibility to provide revised figures in rebuttal, we present the results composed of the mean accuracy and variance in Table 1-Table 6. The revised Figure 5 will be presented in the final version of the paper.
>
> Table 1: Noise experiments on Animal
> |Standard Deviation|$10^0$|$10^1$|$10^2$|$10^3$|$10^4$|$10^5$|$10^6$|$10^7$|
> |-|-|-|-|-|-|-|-|-|
> |ETMC|56.6±0.8|10.4±2.0|4.7±0.3|8.8±1.6|6.3±1.3|4.0±2.2|5.0±0.7|5.4±0.8|
> |mmdynamics|56.7±1.4|34.6±2.0|21.0±0.9|16.8±1.5|15.4±1.6|11.4±2.2|10.2±1.6|10.1±1.6|
> |PDMF|56.9±0.7|11.6±1.6|9.1±0.5|9.1±0.5|9.1±0.5|9.1±0.5|9.1±0.5|9.1±0.5|
> |RCML|56.8±1.2|5.9±1.5|6.9±1.7|6.5±1.8|8.0±0.7|7.9±0.6|8.6±1.5|7.7±1.0|
> |Ours|60.2±1.1|59.6±1.0|58.1±1.2|54.8±1.2|8.9±0.5|8.9±0.5|8.9±0.5|8.5±0.2|
>
> Table 2: Noise experiments on CCV
> |Standard Deviation|$10^0$|$10^1$|$10^2$|$10^3$|$10^4$|$10^5$|$10^6$|$10^7$|
> |-|-|-|-|-|-|-|-|-|
> |ETMC|42.6±1.4|10.4±0.7|10.4±0.7|10.0±0.9|10.4±0.7|7.7±3.1|4.9±1.0|4.9±1.0|
> |mmdynamics|29.6±1.1|24.8±1.2|21.0±1.7|20.3±1.7|19.0±1.4|15.7±1.7|10.3±0.5|10.2±0.5|
> |PDMF|50.5±1.4|17.9±2.9|13.4±2.9|11.8±2.0|10.3±0.6|10.3±0.6|5.4±0.4|5.4±0.4|
> |RCML|42.4±1.7|11.0±1.8|9.7±1.0|10.2±4.9|10.0±1.5|10.0±1.5|10.2±0.8|9.7±1.0|
> |Ours|54.0±1.1|48.9±2.4|48.9±2.8|48.2±2.7|42.7±1.5|42.8±3.2|22.3±2.2|11.1±15.2|
>
> Table 3: Noise experiments on HandWritten
> |Standard Deviation|$10^0$|$10^1$|$10^2$|$10^3$|$10^4$|$10^5$|$10^6$|$10^7$|
> |-|-|-|-|-|-|-|-|-|
> |ETMC|96.8±0.7|94.9±1.7|15.3±8.1|9.3±0.9|13.4±4.1|10.2±2.3|7.2±4.0|9.6±1.2|
> |mmdynamics|98.5±0.5|96.5±1.1|96.2±1.4|80.1±11.7|41.6±13.9|26.5±9.3|27.1±6.6|27.1±6.8|
> |PDMF|98.2±0.7|30.5±8.0|19.1±2.7|19.1±2.7|13.9±3.1|14.0±3.2|8.6±1.4|8.6±1.4|
> |RCML|97.1±0.9|93.2±1.3|12.4±4.9|12.1±3.7|11.8±2.9|11.4±3.2|9.4±1.3|8.8±2.4|
> |Ours|98.9±0.4|98.1±0.6|97.8±0.7|97.4±1.2|97.5±1.0|97.6±1.0|64.4±4.3|42.2±4.2|
>
> Table 4: Noise experiments on Hdigit
> |Standard Deviation|$10^0$|$10^1$|$10^2$|$10^3$|$10^4$|$10^5$|$10^6$|$10^7$|
> |-|-|-|-|-|-|-|-|-|
> |ETMC|90.8±2.1|90.7±2.0|20.6±7.3|10.5±1.2|14.1±6.0|10.6±0.5|11.2±1.5|10.2±0.6|
> |mmdynamics|99.6±0.1|98.7±0.3|93.7±0.8|73.6±7.2|32.1±6.6|22.7±5.4|20.4±3.2|20.5±3.3|
> |PDMF|99.4±0.2|99.6±0.1|19.5±7.5|10.5±0.3|10.5±0.3|10.5±0.3|10.4±0.2|10.4±0.4|
> |RCML|98.3±0.3|94.7±0.9|30.0±3.8|27.9±9.8|28.6±6.6|25.2±8.9|24.2±7.7|10.9±1.8|
> |Ours|99.8±0.1|99.8±0.1|99.6±0.1|97.1±0.5|10.8±2.1|14.7±6.6|10.1±0.6|10.0±0.5|
>
> Table 5: Noise experiments on PIE
> |Standard Deviation|$10^0$|$10^1$|$10^2$|$10^3$|$10^4$|$10^5$|$10^6$|$10^7$|
> |-|-|-|-|-|-|-|-|-|
> |ETMC|90.5±2.3|4.9±1.8|1.2±1.0|1.2±0.9|1.0±0.6|1.4±0.9|1.8±1.4|1.3±1.3|
> |mmdynamics|72.5±3.7|48.3±12.2|32.5±5.8|10.4±3.1|5.3±2.2|3.5±0.8|3.4±1.4|3.6±1.3|
> |PDMF|88.7±3.0|17.8±6.4|3.1±0.3|1.8±0.4|1.8±0.4|1.8±0.4|1.7±0.6|1.7±0.6|
> |RCML|91.8±2.9|10.8±3.2|1.7±1.5|1.7±0.8|2.4±1.4|2.4±1.0|2.5±1.3|2.5±1.5|
> |Ours|93.1±1.7|80.0±6.0|78.3±6.3|2.2±0.6|2.8±1.1|1.8±0.5|0.9±0.8|1.1±0.6|
>
> Table 6: Noise experiments on Scene
> |Standard Deviation|$10^0$|$10^1$|$10^2$|$10^3$|$10^4$|$10^5$|$10^6$|$10^7$|
> |-|-|-|-|-|-|-|-|-|
> |ETMC|66.5±1.8|28.9±4.5|9.0±0.8|6.9±2.2|8.7±1.4|7.6±2.3|7.0±2.7|8.3±3.0|
> |mmdynamics|62.0±2.2|49.6±2.8|50.2±4.0|36.2±1.6|22.3±5.4|16.3±2.4|15.5±2.4|15.5±2.4|
> |PDMF|67.8±1.1|25.1±4.4|11.5±1.8|9.0±0.4|9.0±0.4|9.0±0.4|5.7±0.5|5.7±0.5|
> |RCML|70.0±1.0|28.2±2.1|9.0±0.7|8.3±1.8|9.5±1.6|9.7±2.1|9.4±2.6|10.3±2.1|
> |Ours|81.5±0.7|77.0±2.1|74.9±1.1|9.0±0.4|9.0±0.4|8.5±1.3|8.1±1.5|5.9±1.2|
>
> **W4.**  Thanks for your correction. Our original intention was to use "methods" for indicating the comparative methods in the same row but seemed to create ambiguity. And we will revise the ambiguity in the final version of the paper.

---

> ### Author Response · Authors · 2024-11-18
> **Response to Weakness #5**
>
> **W5.** We find there is only UMDL that encounters the out-of-memory (OOM) issue. The main reason is that the model applies some complex algorithms in the pre-processing phase, such as the singular value decomposition algorithm, which is infeasible for large datasets. And we present the complexity and computational overhead analyses of all the models in Table 7 and Table 8.
>
> Table 7 lists the complexity of all the models. Since the different models may introduce different and complex notations, it is hard to provide detailed complexity for all the models in a unified manner. For convenience, we only adopt $N$ as the data size to calculate the complexity for all the models in Table 7, and we can find that our model has the same level of complexity as some comparative models.
>
> Table 8 provides detailed computational overhead analyses. Because MACs are defined as approximately twice the FLOPs, we only provide MACs in Table 8. Besides, we also list the concrete running time and Parameters for all the models. For a fair comparison, we set the same batch size and training epochs for all the models. From all the statistics from Table 8, we could find that our model is not cumbersome, but is competitive with part advanced models.
>
> Table 7: The complexities among different models.
> |Model      |Complexity|
> |-|-|
> |mmdynamics|$\mathcal{O}(N)$|
> |ETMC      |$\mathcal{O}(N)$|
> |UMDL      |$\mathcal{O}(N^3)$|
> |PDMF      |$\mathcal{O}(N^2)$|
> |RCML      |$\mathcal{O}(N)$|
> |Ours        |$\mathcal{O}(N^2)$|
>
> Table 8: The computational overhead comparisons among different models.
> |Datasets|Metrics|mmdynamics|ETMC|UMDL|PDMF|RCML|Ours|
> |-|-|-|-|-|-|-|-|
> |HandWritten|Running Time|21.911s|35.739s|10057.537s|21.972s|17.064s|23.302s|
> ||MACs|267.871M|3.323M|1273.548M|189.989M|1.661M|1373.655M|
> ||Parameters|1.0543M|0.0066M|5.2389M|1.722M|0.0066M|5.353M|
> |Scene15|Running Time|28.996s|42.596s|31938.726s|45.678s|22.922s|41.651s|
> ||MACs|66.604M|0.457M|132.890M|68.387M|0.457M|617.869M|
> ||Parameters|0.264M|0.002M|20.632M|1.098M|0.002M|3.197M|
> |PIE|Running Time|11.525s|11.233s|2721.368s|9.214s|7.775s|17.499s|
> ||MACs|536.394M|17.739M|248.090M|188.088M|2.217M|996.040M|
> ||Parameters|2.099M|0.070M|1.429M|1.540M|0.070M|4.622M|
> |CCV|Running Time|40.678s|61.986s|29783.373s|69.529s|37.905s|65.490s|
> ||MACs|56.521M|0.307M|125.338M|9.978M|0.307M|263.633M|
> ||Parameters|0.225M|0.001M|46.361M|1.331M|0.001M|3.323M|
> |Animal|Running Time|650.847s|181.911s|52103.473s|221.021s|119.074s|177.946s|
> ||MACs|7643.710M|44.493M|1269.197M|576.355M|44.493M|9360.186M|
> ||Parameters|29.873M|0.174M|141.214M|10.937M|0.174M|37.353M|
> |100Leaves|Running Time|12.697s|16.173s|7386.190s|19.825s|10.078s|44.393s|
> ||MACs|247.370M|4.915M|142.234M|80.740M|4.915M|672.518M|
> ||Parameters|0.971M|0.020M|3.113M|0.814M|0.020M|3.323M|
> |Hdigit|Running Time|57.370s|58.236s|44589.832s|85.163s|41.918s|59.256s|
> ||MACs|506.520M|0.666M|211.558M|42.877M|2.662M|883.593M|
> ||Parameters|1.982M|0.010M|100.825M|2.614M|0.010M|4.245M|
> |YoutubeFace|Running Time|1660.614s|598.449s|-|513.350s|523.801s|702.112s|
> ||MACs|4393.923M|67.456M|-|1466.827M|67.456M|7988.087M|
> ||Parameters|4.328M|0.066M|-|15.840M|0.066M|8.544M|

---

> ### Author Response · Authors · 2024-11-21
> **Response**
>
> Dear Reviewer 1xxb，
>
> Thank you for your time. We would like to kindly remind you that we have responded to your valuable feedback on our submission . If you have any further questions, comments, or concerns, we would be more than happy to address them.
>
> We deeply appreciate the time and effort you have dedicated to reviewing our work and providing constructive feedback, which has been immensely helpful to us.
>
> Please feel free to reach out if there is anything else we can clarify. Thank you again for your contribution to improving our research.
>
> Best regards,
>
> Authors.

---

### Official Review · Reviewer_p7J9 · 2024-11-04

**Soundness:** 3
**Presentation:** 3
**Contribution:** 3
**Rating:** 5
**Confidence:** 4

**Summary:**

The authors propose an expanded boundary using fuzzy set theory to adaptively adjust class boundaries for ambiguous data, enhancing representation tightness and reducing boundary ambiguity. This expanded boundary also preserves more margin for unseen data, improving model generalization. Experiments across diverse datasets show the model’s superiority over state-of-the-art methods.

**Strengths:**

1. The technique appears sound.

2. The presentation is clear and easy to follow.

3. The experimental results are sufficient to demonstrate the effectiveness of the proposed method.

**Weaknesses:**

1. The model includes too many loss terms, which may limit its practicality for real-world applications, since it is often hard to set the hyperparameter to balance the contributions among them in practice.

2. Figure 4 is unclear; I strongly suggest the authors present it in 2D for better readability.  Since there are too many dimensions making it hard to conduct the parameter sensitiveness analysis.

3. Some relevant references are missing [1][2][3], since [1] and [3] also study multi-view classification tasks and [2] focus on multi-view learning from a fuzzy view. The authors should discuss and compare with them.
[1] Multi-view Classification Using Hybrid Fusion and Mutual Distillation, WACV, 2024.
[2] Self-Supervised Information Bottleneck for Deep Multi-View Subspace Clustering, TIP, 2023.
[3] Multi-View Fuzzy Classification With Subspace Clustering and Information Granules, TKDE, 2023.

**Questions:**

Please address the question above.

---

> ### Author Response · Authors · 2024-11-18
> **Response to Weakness #1 and #2**
>
> **W1.** Three to five loss terms are common in multi-view learning since the construction loss and the cross-entropy loss are the common terms in the existing works [1-4]. Except for these terms, we only introduce two losses, namely the instance-scale alignment loss and the prototype-scale alignment loss for addressing the feature heterogeneity and the information redundancy. Besides, we simply set the penalty coefficients of the construction loss and the cross-entropy loss as 1 to alleviate the burden of hyperparameter tuning, which is also more convenient for attending to the effects of the proposed modules.
>
> [1] Reliable conflictive multi-view learning. AAAI 2024.
>
> [2] Triple-granularity contrastive learning for deep multi-view subspace clustering. ACM MM 2023.
>
> [3] Trusted multi-view classification with dynamic evidential fusion. IEEE TPAMI, 2023.
>
> [4] Self-supervised information bottleneck for deep multi-view subspace clustering. IEEE TIP 2023.
>
> **W2.** Thanks for your suggestion. We drew Figure 4 in 3D by referring to [1], which might lack consideration for better readability. Following your suggestion, we now provide the elaborate results of the parameter sensitiveness analyses in Table 1 - Table 4 since we are unable to provide the revised 2D figures directly in the rebuttal. From the results, we observe that model performance initially improves and then declines as the hyperparameters increase. This indicates that all the hyperparameters contribute to model performance, validating the role of each loss term and module. Specifically, from Tables 1 and 2, we find that the model performs best when both $\alpha$ and $\beta$ are properly tuned. This suggests that the instance-scale alignment and prototype-scale alignment modules work together to enhance overall model performance.
>
> Table 1: Hyperparameter analyses for penalty coefficients on HandWritten.
> |Hyperparameters|$\beta=0.1$|$\beta=0.25$|$\beta=0.5$|$\beta=0.75$|$\beta=1.0$|
> |-|-|-|-|-|-|
> |$\alpha=0.1$|96.5±0.6|97.5±0.4|97.9±0.6|97.9±0.4|97.7±0.5|
> |$\alpha=0.25$|97.2±0.4|97.8±0.4|98.3±0.5|98.4±0.5|98.0±0.6|
> |$\alpha=0.5$|97.4±0.4|98.2±0.4|98.8±0.5|98.6±0.5|98.3±0.6|
> |$\alpha=0.75$|97.2±0.6|97.7±0.6|98.6±0.6|98.3±0.4|98.5±0.6|
> |$\alpha=1.0$|97.2±0.7|98.0±0.7|98.2±0.4|98.4±0.7|98.2±0.4|
>
> Table 2: Hyperparameter analyses for penalty coefficients on Hdigit.
> |Hyperparameters|$\beta=0.1$|$\beta=0.25$|$\beta=0.5$|$\beta=0.75$|$\beta=1.0$|
> |-|-|-|-|-|-|
> |$\alpha=0.1$|98.5±0.2|98.5±0.2|98.7±0.2|98.6±0.2|98.6±0.2|
> |$\alpha=0.25$|98.7±0.2|99.0±0.2|99.3±0.2|99.4±0.1|99.4±0.1|
> |$\alpha=0.5$|99.2±0.2|99.5±0.2|99.7±0.2|99.7±0.2|99.5±0.2|
> |$\alpha=0.75$|99.2±0.2|99.3±0.2|99.6±0.2|99.6±0.2|99.5±0.2|
> |$\alpha=1.0$|99.0±0.2|99.2±0.2|99.3±0.2|99.4±0.2|99.1±0.2|
>
> Table 3: Hyperparameter analyses for temperature coefficients on HandWritten.
> |Hyperparameters|$\tau_{pro}=0.5$|$\tau_{pro}=0.75$|$\tau_{pro}=1.0$|$\tau_{pro}=1.2$|$\tau_{pro}=1.5$|
> |-|-|-|-|-|-|
> |$\tau_{ins}=0.5$|98.0±0.5|98.3±0.4|98.1±0.4|98.6±0.4|98.6±0.5|
> |$\tau_{ins}=0.75$|97.8±0.5|98.0±0.5|98.3±0.5|98.5±0.5|98.3±0.5|
> |$\tau_{ins}=1.0$|98.3±0.5|98.0±0.6|98.5±0.5|98.8±0.4|98.5±0.5|
> |$\tau_{ins}=1.5$|98.1±0.6|98.3±0.6|98.5±0.4|98.5±0.6|98.5±0.7|
> |$\tau_{ins}=2.0$|98.3±0.4|98.6±0.6|98.8±0.7|98.8±0.6|98.5±0.4|
>
> Table 4: Hyperparameter analyses for temperature coefficients on Hdigit.
> |Hyperparameters|$\tau_{pro}=0.5$|$\tau_{pro}=0.75$|$\tau_{pro}=1.0$|$\tau_{pro}=1.2$|$\tau_{pro}=1.5$|
> |-|-|-|-|-|-|
> |$\tau_{ins}=0.5$|98.8±0.1|99.0±0.2|99.1±0.2|99.1±0.2|99.0±0.2|
> |$\tau_{ins}=0.75$|99.0±0.2|99.2±0.2|99.3±0.2|99.4±0.1|99.1±0.1|
> |$\tau_{ins}=1.0$|99.2±0.2|99.5±0.2|99.7±0.2|99.7±0.2|99.7±0.2|
> |$\tau_{ins}=1.5$|99.2±0.2|99.7±0.2|99.7±0.2|99.7±0.2|99.5±0.2|
> |$\tau_{ins}=2.0$|99.2±0.2|99.5±0.2|99.7±0.2|99.7±0.2|99.6±0.2|
>
> [1] "Triple-granularity contrastive learning for deep multi-view subspace clustering." ACM MM 2023.

---

> ### Author Response · Authors · 2024-11-18
> **Response to Weakness #3**
>
> **W3.** We have tried our best to find the source codes of the above papers for discussion and comparison, such as searching in GitHub and contacting with the authors. However, we only found source codes of [1] and [3]. Here, we provide the discussions about these references in the following, and the comparative results on classification are shown in Table 5.
>
> Discussion and comparison:
>
> (1) MV-HFMD [1] delves into feature fusion and incorporates a novel fusion scheme and mutual distillation to repurpose off-the-shelf neural networks for multi-view classification. From the results in Table 3, MV-HFMD shows a very outstanding performance, which implies the significance of effective feature fusion in multi-view learning. However, it may neglect to address the feature heterogeneity that plays an important in facilitating the feature fusion, so the model performance is somewhat limited when compared with our model.
>
> (2) SIBMSC [2] aims at learning the minimal sufficient latent representation for each view to obtain common information with the information bottleneck theory. It mainly focuses on the inter-view common information but does not consider the inter-cluster or inter-class information. However, our model focuses on both information. Additionally, we need to point out that this paper may not focus on multi-view learning from a fuzzy view.
>
> (3) IPMVSC [3] proposes an anchor and graph subspace clustering strategy to discover and represent the actual latent data distribution for each view separately, and constructs a multi-view fuzzy classification model inherited from the natural interpretability of fuzzy rule-based systems. Therefore, this work focuses on using fuzzy rules to realize an interpretable classifier. However, our model attends to the feature-level issues and applies the fuzzy set theory in the representation space. Besides, from the results, we could also find IPMVSC does not work well on large datasets since the model makes it difficult to select the appropriate hyperparameters to balance the overfitting problem and model performance as the author stated [3], but our model works well on both small and large datasets.
>
>
> Table 5: Comparisons of classification results for supplemented comparative models and our model.
> |Datasets|Metrics|MV-HFMD [1]|IPMVSC [3]|Ours|
> |-|-|-|-|-|
> |HandWritten|Acc|98.15±0.58|98.70±0.41|**98.85±0.42**|
> ||Precision|98.18±0.51|98.69±0.43|**98.87±0.39**|
> ||Recall|98.15±0.56|98.72±0.39|**98.87±0.41**|
> ||Macro F1|98.14±0.53|98.68±0.41|**98.85±0.41**|
> |Scene15|Acc|80.65±1.19|71.84±1.41|**81.52±0.65**|
> ||Precision|80.58±1.57|73.82±0.66|**80.99±0.56**|
> ||Recall|80.33±1.28|70.51±0.82|**80.75±0.69**|
> ||Macro F1|80.00±1.44|69.13±0.87|**80.48±0.66**|
> |PIE|Acc|86.76±2.41|91.91±2.03|**93.10±1.72**|
> ||Precision|87.65±2.74|81.72±2.14|**92.82±2.13**|
> ||Recall|88.86±2.41|82.11±1.84|**93.63±1.77**|
> ||Macro F1|83.75±6.11|80.54±2.54|**92.12±2.07**|
> |CCV|Acc|53.36±0.86|42.59±1.83|**53.97±1.13**|
> ||Precision|51.08±1.12|48.32±1.99|**53.00±1.26**|
> ||Recall|50.53±0.83|35.47±1.27|**50.62±0.99**|
> ||Macro F1|50.39±0.89|34.30±1.51|**50.96±0.93**|
> |Animal|Acc|59.58±0.67|39.51±1.93|**60.16±1.10**|
> ||Precision|56.47±0.55|37.34±3.88|**56.74±1.31**|
> ||Recall|53.68±0.40|33.31±2.46|**53.76±0.81**|
> ||Macro F1|54.01±0.46|33.33±2.72|**54.10±0.81**|
> |100Leaves|Acc|98.28±0.16|66.06±4.14|**98.53±1.06**|
> ||Precision|98.36±0.01|70.19±3.42|**98.50±1.03**|
> ||Recall|98.93±0.24|68.97±2.03|**98.96±0.74**|
> ||Macro F1|98.40±0.10|63.95±2.67|**98.45±1.04**|
> |Hdigit|Acc|87.40±2.0|97.84±0.28|**99.77±0.10**|
> ||Precision|88.90±1.67|97.83±0.28|**99.77±0.10**|
> ||Recall|87.52±1.99|97.86±0.26|**99.77±0.10**|
> ||Macro F1|86.49±2.26|97.83±0.27|**99.77±0.10**|
> |YoutubeFace|Acc|83.42±0.77|28.18±0.32|**87.07±0.15**|
> ||Precision|85.62±1.07|29.92±3.97|**89.46±0.42**|
> ||Recall|82.83±0.81|5.43±0.28|**86.17±0.31**|
> ||Macro F1|83.94±0.75|5.15±0.35|**87.71±0.14**|

---

> ### Author Response · Authors · 2024-11-21
> **Response**
>
> Dear Reviewer p7J9，
>
> Thank you for your time. We would like to kindly remind you that we have responded to your valuable feedback on our submission . If you have any further questions, comments, or concerns, we would be more than happy to address them.
>
> We deeply appreciate the time and effort you have dedicated to reviewing our work and providing constructive feedback, which has been immensely helpful to us.
>
> Please feel free to reach out if there is anything else we can clarify. Thank you again for your contribution to improving our research.
>
> Best regards,
>
> Authors.

---

### Author Response · Authors · 2024-11-28

Dear Reviewers,

Following the discussion phase, we have updated our manuscript to address the points raised in our earlier exchanges. To assist with your review, we have highlighted the revised sections directly in the document for your convenience.

We sincerely appreciate your time and thoughtful feedback. Please let us know if there is anything further we can clarify.

Best regards,

Authors

---

### Author Response · Authors · 2024-11-30

Dear Reviewers,

Thank you again for your valuable feedback. We have carefully addressed your comments in the rebuttal and revised the manuscript accordingly.

We kindly invite you to review the updates, and please let us know if you have any further questions or suggestions. Your time and insights are greatly appreciated.

Best regards,

Authors

---

### Author Response · Authors · 2024-12-02

Dear Reviewers,
﻿

We hope this message finds you well. We have addressed your comments and revised the manuscript accordingly, submitting both the updated version and our rebuttal for your review.
﻿

With the rebuttal deadline approaching, we would greatly appreciate it if you could review the changes and provide your feedback. Please do not hesitate to reach out if there are any questions.
﻿

Thank you for your valuable insights and time.
﻿

Best regards,

Authors

---

### Meta-Review · Area_Chair_2dN8 · 2024-12-24

**Metareview:**

The paper presents the MAMC model to address feature heterogeneity in multi-view data through view-specific auto-encoders and a multi-scale alignment module. The paper is well-structured, with clear motivation, detailed design, and theoretical proof. Despite these merits, reviewers raised several major concerns regarding computational complexity and efficiency, lacking comparisons with recently published methods. Other minors include clarity of explanations for feature heterogeneity and symbol definitions. After the discussion period, the authors were able to implement all the comments in the revised manual, and AC was convinced by their efforts and new content and thus recommended accepting this paper.

**Additional Comments On Reviewer Discussion:**

Two reviewers were satisfied with the authors' responses, while the other two did not provide a response after the discussion period. Despite the partial responses and confirmation from the reviewers, most of the reviewers chose to maintain their original rating. Since the authors managed to address all comments and implement them in the revised version, the AC was convinced by the authors' contributions and improvement.

---

### Decision · Program_Chairs · 2025-01-22

Accept (Poster)